# REVISITING VECTOR-QUANTIZATION FOR BLIND IMAGE RESTORATION

## ABSTRACT

Vector-Quantization (VQ) generative models are widely used to learn a high-quality (HQ) codebook and a decoder as powerful generative priors for blind image restoration (BIR). In this paper, we revisit the key VQ process in VQ-based BIR methods, and provide three close observations on the side effects of VQ for code index prediction: 1) confining the representational capability of HQ codebook, 2) being error-prone on code index prediction, and 3) under-valuing the low-quality (LQ) feature for BIR. These observations motivate us to replace discrete VQ selection by continuous feature transformation from input LQ image to output HQ image with the HQ codebook. To this end, in this paper, we propose a new **S**elf-**in**-**C**ross-**A**ttention (**SinCA**) module to augment the HQ codebook with the LQ feature of input LQ image and perform cross-attention between LQ feature and input-augmented codebook. In this way, our SinCA extends the representational capability of the HQ codebook and effectively leverages the self-expressiveness property of input LQ image. Experiments on four typical VQ-based BIR methods demonstrate that, by replacing the VQ process with transformers using our SinCA, they achieve better quantitative and qualitative performance on blind image super-resolution and blind face restoration. The code will be publicly released.

## 1 INTRODUCTION

Blind image restoration (BIR) aims to recover high-quality (HQ) images from the corresponding low-quality (LQ) images affected by complex degradation (Wang et al., 2021c; 2023b; Zhang et al., 2021). This ill-posed problem has been addressed by many generative models (Wang et al., 2021b; Chen et al., 2022; Lin et al., 2023; Wang et al., 2023a), under the architectures of VAEs (Kingma & Welling, 2013; Rezende et al., 2014), GANs (Goodfellow et al., 2014; Karras et al., 2019), or diffusion models (Ho et al., 2020; Song et al., 2020). Recently, with successes in applications like DALL·E (Ramesh et al., 2021), Vector-Quantization (VQ) based discrete generative models like VQVAE (Van Den Oord et al., 2017) or VQGAN (Esser et al., 2021) have been increasingly adopted in many BIR methods (Chen et al., 2022; Zhou et al., 2022; Liu et al., 2023a; TSAI et al., 2024) as robust backbones against diverse image degradations.

Current VQ-based BIR methods typically follow a multi-stage training scheme. First, an encoder-decoder model and a discrete codebook are learned to reconstruct HQ images using VQGAN (Esser et al., 2021). The encoder is then fine-tuned to restore LQ images, during which the VQ process replaces each pixel-wise LQ feature vector with a selected code item from the HQ codebook. Though allowing for useful information recovery from the HQ codebook, the VQ process also brings three notable side effects to VQ-based blind restoration methods. Firstly, the representational range of VQ process is confined to the finite set of HQ codebook items. Secondly, two main VQ strategies, *i.e.*, nearest-neighbor feature matching (Fig. 1 (a)) and transformer-based prediction (Fig. 1 (c)), are error-prone on selecting code items for BIR. Thirdly, the VQ process undervalues the essential role of LQ features for blind image restoration, since it simply replaces LQ features by HQ code items.

Considering the side effects of VQ process mentioned above, a natural question raises: is it feasible to replace the vulnerable discrete VQ process by continuous transformation from LQ features to HQ ones with the HQ codebook? In this paper, we provide positive feedback to the above question by implementing continuous feature transformation via cross-attention between LQ feature and HQ codebook. Specifically, the cross-attention computes the attention map by using the LQ feature as the

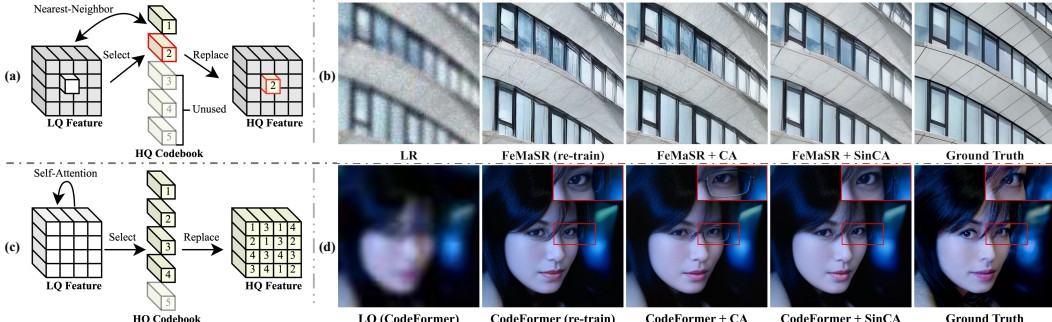

Figure 1: **Left**: Illustration of two VQ strategies. (a) Nearest-neighbor feature matching selects the closest (via a distance metric) HQ code item for each LQ feature vector. (c) Transformer-based code index prediction learns a transformer to predict the HQ code index for each LQ feature vector. **Right**: Effectiveness of replacing discrete VQ process by continuous feature transformation. (b) "FeMaSR+CA" or "FeMaSR+SinCA" denotes a FeMaSR variant replacing nearest-neighbor feature matching by cross-attention or SinCA, respectively. (d) "CodeFormer+CA" or "CodeFormer+SinCA" denotes a CodeFormer variant replacing the self-attention for index prediction by cross-attention or SinCA, respectively. CA generates a faked glass when restoring the LQ face image.

query and the HQ codebook as the key, highlighting the most relevant HQ code items for each LQ feature vector. The HQ feature is obtained by multiplying the attention map with the HQ codebook as the value. Here, each HQ feature vector is obtained by a weighted combination of HQ code items, where the weights stands for adaptive correlation between the corresponding LQ feature vector and the HQ code items. In this way, the representation range of HQ codebook extends from a set of finite code items in VQ to a simplex space spanned by HQ code items with infinite combinations.

Despite its effectiveness, naive cross-attention taking only HQ codebook as the key and value would fails to capture specific characteristics in diverse LQ images. The reason is that the HQ codebook are learned from external HQ images. To address this issue, we propose a new Self-in-Cross-Attention (SinCA) module to conduct cross-attention between the LQ feature of input LQ image and an input-augmented codebook consisting of the HQ codebook and the feature of input LQ image. To be specific, our SinCA contains two collaborative parts: cross-part and self-part. On one hand, the cross-part exploits the adaptive correlation information between input LQ feature and HQ codebook. On the other, the self-part excavates the self-expressiveness property (Elhamifar & Vidal, 2013) of each input LQ image, which is very useful to boost BIR performance, as shown in Figs. 1 (b) and (d).

In summary, the contributions of this work are three-fold:

- We revisit the key VQ process of VQ-based BIR methods, and provide three key observations on the side-effects of discrete VQ process on code index prediction.

- Our observations raise the necessity to replace discrete VQ process by continuous feature learning. To this end, we propose a **S**elf-**in**-**C**ross-**A**ttention (SinCA) module with augmented codebook to simultaneously exploit useful information from input LQ image and HQ codebook.

- Experiments demonstrate that, by replacing the VQ process with a standard transformer using our SinCA, four typical VQ-based BIR methods achieve better performance on blind image super-resolution and blind face restoration. Ablation studies validate the effectiveness of our SinCA.

## 2 RELATED WORK

**Blind Image Restoration** (BIR) aims to recover high-quality (HQ) images from the corresponding low-quality (LQ) images with unknown degradation. Early methods (Gu et al., 2019; Huang et al., 2020; Shocher et al., 2018; Zhang et al., 2018a) mainly exploit the effectiveness of CNNs to restore LQ images. To tackle the complex degradation, many different generative priors (Chen et al., 2018; Shen et al., 2018; Yu et al., 2018; Zhu et al., 2022) have been utilized to achieve robust restoration performance. Among these priors, GAN priors (Goodfellow et al., 2014; Karras et al., 2019) are widely adopted by BIR methods (Pan et al., 2020; 2021; Tao Yang & Zhang, 2021; Wang et al., 2021b; 2022a). For example, GFPGAN (Wang et al., 2021b) and GPEN (Tao Yang & Zhang, 2021) employed a pre-trained StyleGAN2 (Karras et al., 2020) in a U-shaped decoder network for face

image restoration. However, GANs are prone to generate unrealistic textures due to the inherent difficulty on distinguishing similar patterns (Chen et al., 2022). Recently, diffusion generative priors (Ho et al., 2020; Song et al., 2020) have also been exploited by many BIR methods (Wang et al., 2023a; Lin et al., 2023; Yang et al., 2023). Despite the impressive progress, these methods are usually not robust to severe image degradations (Zhou et al., 2022).

Benefited by the power of vector-quantization (VQ) models (Van Den Oord et al., 2017; Esser et al., 2021) in image generation, VQ-based BIR methods (Zhao et al., 2022; Chen et al., 2022; Zhou et al., 2022; Liu et al., 2023a; Wang et al., 2023b) have been developed to utilize discrete HQ codebook priors. In this work, we examine the side effects of discrete VQ process on feature matching and replace it with our Self-in-Cross-Attention (SinCA) for continuous feature learning.

**VQ-based Generative Models** learn discrete codebook priors of images in latent space. This idea is first introduced in VQVAE (Van Den Oord et al., 2017) and further enhanced by VQGAN (Esser et al., 2021) with better perceptual quality induced by learning codebook priors and autoregressive transformer (Vaswani et al., 2017). Built upon VQVAE and VQGAN, recent VQ-based image generation methods (Cao et al., 2023; Chang et al., 2022; Lee et al., 2022; Yu et al., 2022; Zhang et al., 2023; Zheng et al., 2022) primarily focus on improving the quantization and token generation processes. For example, MaskGIT (Chang et al., 2022) utilized a bidirectional transformer (Kenton & Toutanova, 2019) to simultaneously predict all the image tokens.

VQ-based generative priors have also been adopted by many methods for face restoration (Zhou et al., 2022; Gu et al., 2022; TSAI et al., 2024; Wang et al., 2023b; Zhao et al., 2022) and image super-resolution (Chen et al., 2022; Liu et al., 2023a; Wu et al., 2023; Liu et al., 2023b). In particular, the methods of FeMaSR (Chen et al., 2022), AdaCode (Liu et al., 2023a), and RestoreFormer++ (Wang et al., 2023b) performed codebook selection via nearest-neighbor (NN) feature matching. Code-Former (Zhou et al., 2022) predicted the indices of code items using transformers. DAEFR (TSAI et al., 2024) used an extra HQ encoder as the prior to bridge the domain gap between LQ and HQ images. AdaCode learned five categories of HQ codebooks with a weight predictor to effectively restore the LQ images. In this paper, we propose to replace the discrete VQ process by continuous transformation from LQ feature to HQ ones with the HQ codebook for VQ-based BIR.

**Vision Transformer** (Dosovitskiy et al., 2021) has inspired great progress in computer vision tasks (Wang et al., 2021a; Carion et al., 2020). It extends the idea of self-attention (Vaswani et al., 2017) by taking a sequence of image patches as input tokens. SwinIR (Liang et al., 2021) performed self-attention on shifted local windows (Liu et al., 2021) and transmitted information between them. Restormer (Zamir et al., 2022) exploited self-attention across feature channels for efficient image restoration. Cross-attention is also developed to mix the information from two different inputs (Chen et al., 2021). It is applied in RestoreFormer (Wang et al., 2022b; 2023b) to fuse the LQ and HQ features for blind face restoration. In this paper, we propose a Self-in-Cross-Attention module to collaboratively perform self-attention of LQ feature and cross-attention between LQ feature and HQ codebook for feature learning in VQ-based BIR methods.

## 3 PRELIMINARY

**Vector Quantization** (VQ) is a classical quantization technology originally developed for signal compression (Linde et al., 1980). With VQ, VQVAE (Van Den Oord et al., 2017) learns an encoder $\mathbf{E}$, a decoder $\mathbf{D}$, and a discrete visual codebook $\mathbf{C} = [\mathbf{c}_1, ..., \mathbf{c}_B]^\top \in \mathbb{R}^{B \times d}$ of images in latent space with a deep neural network. Given an input image $\mathbf{x}$, the encoder $\mathbf{E}$ extracts its latent feature as $\mathbf{z} = \mathbf{E}(\mathbf{x}) \in \mathbb{R}^{h \times w \times d}$, which is then quantized by replacing each of its feature vector $\mathbf{z}_i$ ($i = 1, ..., hw$) with the corresponding nearest code item in codebook $\mathbf{C}$, as follows:

$$\hat{\mathbf{z}}_i = \mathbf{c}_{k_i}, \text{ where } k_i = \underset{j \in \{1, ..., B\}}{\arg\min} \|\mathbf{z}_i - \mathbf{c}_j\|_2. \tag{1}$$

The quantized latent feature $\hat{\mathbf{z}}$ including all replaced code items $\{\hat{\mathbf{z}}_i\}_{i=1}^{hw}$ is fed into the decoder $\mathbf{D}$ to output the reconstructed image. For model training, VQVAE utilizes three loss functions, *i.e.*, a reconstruction loss $\mathcal{L}_{\text{rec}}$ to minimize the distance between the output and the target image $\mathbf{y}$, a codebook loss $\mathcal{L}_{\text{code}}$ and a commitment loss $\mathcal{L}_{\text{com}}$ with a weighting factor $\beta$. Denoting sg($\cdot$) as the stop-gradient operator (Van Den Oord et al., 2017), the overall objective function $\mathcal{L}_{\text{total}}$ is as follows:

$$\mathcal{L}_{\text{total}} = \underbrace{\|\mathbf{y} - \mathbf{D}(\hat{\mathbf{z}})\|_2^2}_{\mathcal{L}_{\text{rec}}} + \underbrace{\|\text{sg}(\mathbf{E}(\mathbf{x})) - \hat{\mathbf{z}}\|_2^2}_{\mathcal{L}_{\text{code}}} + \beta \underbrace{\|\text{sg}(\hat{\mathbf{z}}) - \mathbf{E}(\mathbf{x})\|_2^2}_{\mathcal{L}_{\text{com}}}. \tag{2}$$

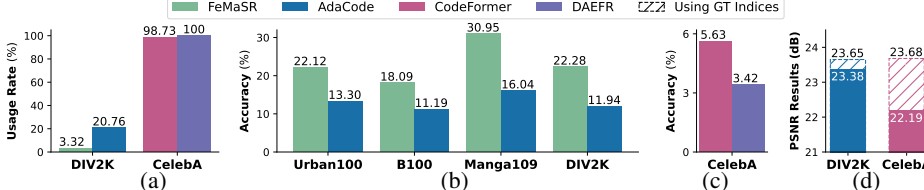

Figure 2: (a) Usage rates of codebook in FeMaSR, AdaCode, CodeFormer and DAEFR. (b) Prediction accuracy of code index in FeMaSR and AdaCode for ×2 task on different test sets. (c) Prediction accuracy of code index in CodeFormer and DAEFR on synthetic CelebA-Test set. (d) Inaccurate index prediction brings performance drop. AdaCode and CodeFormer achieve better results when using ground-truth (GT) indices (Sec.§4.2).

Since quantization is non-differentiable, VQVAE adopts straight-through gradient estimator (Huh et al., 2023) to back-propagate the gradients of the reconstruction loss $\mathcal{L}_{\text{rec}}$ from decoder to encoder. VQGAN (Esser et al., 2021) further improves VQVAE by extraly utilizing an adversarial loss (Goodfellow et al., 2014) and a perceptual loss (Johnson et al., 2016) for better reconstruction quality. These VQ-based generative models have inspired many VQ-based blind image restoration methods (Chen et al., 2022; Zhou et al., 2022; Liu et al., 2023a; Wang et al., 2023b; TSAI et al., 2024).

**VQ-based Blind Image Restoration (BIR)** methods mainly leverage the learned codebook and decoder as a high-quality (HQ) prior robust to diverse degradation. The training pipeline of these methods can be generally divided into two stages with different goals. The first **Prior Learning** stage aims to reconstruct the HQ image $\mathbf{x}^h$ by learning an HQ encoder $\mathbf{E}_h$, an HQ decoder $\mathbf{D}_h$, and an HQ codebook $\mathbf{C}$. The second **Restoration** stage is to restore the low-quality (LQ) images along with the learned HQ prior, *i.e.*, the HQ decoder $\mathbf{D}_h$ and HQ codebook $\mathbf{C}$. To this end, these methods learn an LQ encoder $\mathbf{E}_l$ (initialized from the HQ encoder $\mathbf{E}_h$) to extract from the LQ image $\mathbf{x}^l$ its latent feature $\mathbf{z}^l = \mathbf{E}_l(\mathbf{x}^l)$. Each vector $\mathbf{z}_i^l$ in $\mathbf{z}^l$ is replaced by a predicted code item in HQ codebook $\mathbf{C}$ via a VQ process, usually implemented by nearest-neighbor feature matching (Chen et al., 2022; Liu et al., 2023a) or code index prediction (Zhou et al., 2022; TSAI et al., 2024). The quantized HQ feature $\hat{\mathbf{z}}$ is fed into the HQ decoder $\mathbf{D}_h$ to recover the HQ image $\mathbf{x}^h$. Besides the two stages, many VQ-based BIR methods (Zhou et al., 2022; Wang et al., 2023b) further fuse the LQ feature from encoder and the HQ feature from decoder to trade-off the restoration fidelity and quality.

## 4 OBSERVATIONS ON VQ-BASED BLIND IMAGE RESTORATION METHODS

Despite promising performance, current VQ-based methods (Chen et al., 2022; Zhou et al., 2022; Wang et al., 2022b; 2023b; Liu et al., 2023a; TSAI et al., 2024) rarely discuss the potential side effects of the essential VQ process for blind image restoration (BIR). In this section, we provide three close observations on the VQ process in the second **Restoration** stage of VQ-based BIR methods.

### 4.1 OBSERVATION 1: VQ CONFINES THE CODEBOOK'S REPRESENTATIONAL CAPABILITY

The high-quality (HQ) codebook serves as an expressive generative prior for VQ-based BIR (Chen et al., 2022; Zhou et al., 2022; Liu et al., 2023a; TSAI et al., 2024). VQ performs one-hot code selection to replace each low-quality (LQ) feature vector by a single HQ code item from the HQ codebook. This, however, confines the representation range of HQ codebook to a finite set of code items. This limitation would be further amplified by low codebook usage rates of VQ-based BIR methods. As illustrated in Fig. 2, though the codebook usage rate of CodeFormer (Zhou et al., 2022) and DAEFR (TSAI et al., 2024) are 98.73% and 100%, respectively, for blind face restoration on 3,000 face images from CelebA-Test (Karras et al., 2018)). FeMaSR (Chen et al., 2022) and AdaCode (Liu et al., 2023a) only used 3.32% and 20.76% of the HQ codebook vectors, respectively, for ×2 blind image super-resolution on the DIV2K validation set.

Since the representational capability of VQ process is confined by the discrete code selection of HQ codebook in VQ-based BIR methods, it is necessary to develop alternative solutions that can well utilize HQ codebook and expand the representational range of HQ codebook on diverse LQ images.

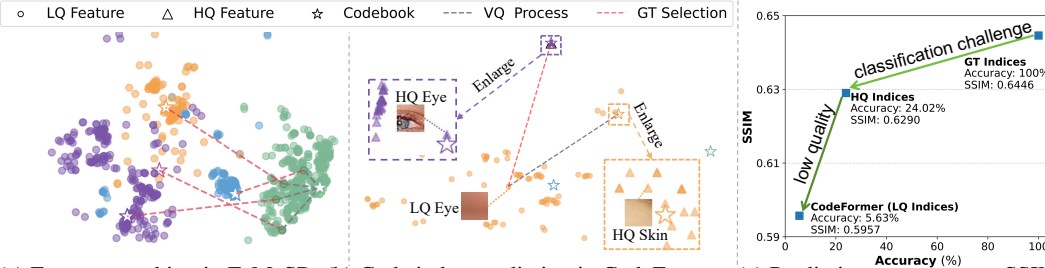

(a) Feature matching in FeMaSR. (b) Code index prediction in CodeFormer. (c) Prediction accuracy *v.s.* SSIM

Figure 3: **T-SNE visualization of VQ process** in FeMaSR (a) and CodeFormer (b). Different code items in HQ codebook are marked by "☆" in different colors. The color of LQ feature vector marked by "○" or the HQ feature vector marked by "△" is the same with the codebook item they select in VQ process. Gray dashed lines "- -" connects the LQ feature vector ○ and its selected codebook item "☆" ("VQ Process"). Red dash lines "- -" connects the LQ feature vector ○ and the codebook item "☆" selected by the corresponding HQ feature vector using nearest-neighbor (NN) feature matching in the **Prior Learning** Stage ("GT Selection"). (a) NN feature matching on LQ feature vectors are inconsistent with "GT Selection". For a given LQ feature vector, the codebook item selected by NN feature matching (gray dash line, "- -") is quite different from the corresponding "GT Selection" ( Red dashed line, "- -"). (b) Transformer for code index prediction is not robust to image degradation. In a degraded LQ image, an "LQ Eye" patch looks like skin area and selects the code item represented by many "HQ Skin" patches. The regions marked by purple dashed box and orange dashed box are enlarged for better view ("Enlarge"). (c) Prediction accuracy of code indices by transformer and SSIM results achieved by CodeFormer using "LQ Indices" predicted on LQ images, "HQ Indices" predicted on HQ images, or "GT Indices" defined in §4.2.

## 4.2 OBSERVATION 2: THE VQ PROCESS IS ERROR-PRONE ON LQ FEATURES

Current VQ-based BIR methods mainly adopt two VQ strategies for code index prediction: 1) nearest-neighbor feature matching independently selects a nearest code item from HQ codebook for each LQ feature vector (Chen et al., 2022; Liu et al., 2023a) and 2) learning a transformer to exploit global correlations of the input LQ image for code index prediction (Zhou et al., 2022; TSAI et al., 2024). However, both strategies suffer from inaccurate code selection. To illustrate this point, we evaluate mainstream VQ-based BIR methods on the prediction accuracy of code index, which refers to the percentage that, the number of indices predicted from LQ feature vectors equal their ground-truth (GT) indices. The GT indices are obtained through nearest-neighbor feature matching in Eqn. (1) using the corresponding HQ feature vectors $\mathbf{z}^h = \mathbf{E}_h(\mathbf{x}^h)$ (Zhou et al., 2022). In Figs. 2 (b) and (c), we visualize the prediction accuracies of four typical VQ-based BIR methods. The accuracies of FeMaSR (Chen et al., 2022) and AdaCode (Liu et al., 2023a) are at most $30.95\%$ on different test sets, while those of CodeFormer (Zhou et al., 2022) and DAEFR (TSAI et al., 2024) are $5.63\%$ and $3.42\%$, respectively. As shown in Fig. 2 (d), AdaCode (Liu et al., 2023a) and CodeFormer (Zhou et al., 2022) achieve higher PSNR results when using GT code indices. This demonstrates that low accuracy of code index prediction degrades the performance of VQ-based BIR methods.

The low prediction accuracy is mainly attributed to the quality degradation of LQ images, as shown in Figs. 3 (a) and (b). Figs. 3 (c) also shows that, using HQ images for index prediction in CodeFormer ("HQ Indices") increases the accuracy from $5.63\%$ to $24.02\%$ with clear improvements on SSIM. Furthermore, learning a transformer to predict code indices (Zhou et al., 2022) casts this problem as a classification task. However, this is error-prone since CodeFormer has $1024^{256} \approx 10^{768}$ possible prediction choices even on a $16 \times 16$ LQ feature with an HQ codebook of 1024 items. As shown in Fig. 2 (c), both the image quality degradation and classification challenge conspire to the clear drops in prediction accuracy of code index and SSIM results of CodeFormer. Besides, the VQ process in (Chen et al., 2022; Liu et al., 2023a) also brings gradient estimation errors when back-propagate the gradients from decoder to encoder (Huh et al., 2023), as described in Sec. 3. Thus, it is potentially meaningful to replace the discrete VQ process by alternative solutions that are feasible to perform HQ feature learning while avoiding error-prone index prediction.

## 4.3 OBSERVATION 3: LQ FEATURE IS IMPORTANT FOR BIR, BUT UNDERVALUED IN VQ

In VQ-based BIR methods (Zhou et al., 2022), the VQ process directly replace the LQ features by selected HQ code items to retrieve high-quality image information. However, this fails to establish a

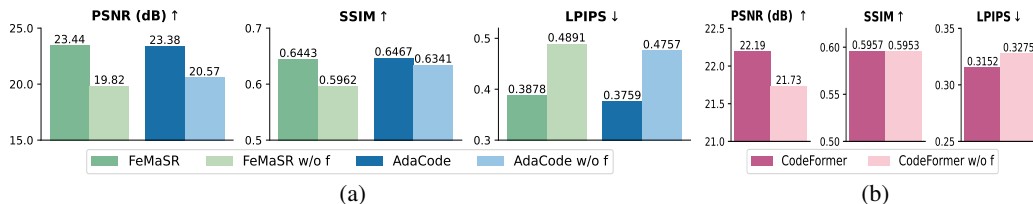

Figure 4: **Importance of LQ feature for BIR**. (a) Quantitative results of FeMaSR and AdaCode *w* or *w/o* feature fusion for ×2 blind super-resolution on DIV2K validation set. (b) Quantitative results of CodeFormer *w* or *w/o* feature fusion on synthetic CelebA-Test set.

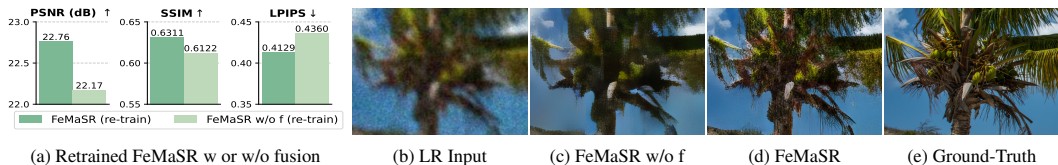

(a) Retrained FeMaSR w or w/o fusion      (b) LR Input      (c) FeMaSR w/o f      (d) FeMaSR      (e) Ground-Truth

Figure 5: **Comparison of retrained FeMaSR *with (w)* or *without (w/o)* feature fusion module**. (a) Quantitative results on DIV2K-validation set. Given an LR image (b), the retrained FeMaSR *w/o* feature fusion (c) loses many texture details when compared to the FeMaSR *w* feature fusion (d).

direct connection between the LQ features and the final restoration performance. To alleviate this issue, many VQ-based BIR methods (Chen et al., 2022; Zhou et al., 2022) further fuse the LQ feature from encoder and the HQ features from decoder to enhance the restoration performance. To study the role of LQ features, we perform experiments on the released models of FeMaSR, AdaCode, and CodeFormer. We remove the corresponding feature fusion module from these models and denote the corresponding variants as "FeMaSR w/o f", "AdaCode w/o f", and "CodeFormer w/o f", respectively. As shown in Figs. 4 (a) and (b), it is not surprising to observe a huge drop of these variants on BIR. We also retrained the restoration stage of FeMaSR and "FeMaSR w/o f". The results in Fig. 5 show that the retrained variant "FeMaSR w/o f" still suffers from clear performance drop. These results validate that the feature of input LQ image is essential to final BIR performance.

Despite their efforts to preserve LQ information, these VQ-based BIR methods are still constrained by the VQ bottleneck. As the LQ feature is only used for the prediction of code index, regardless of how informative the LQ feature is, the information about the LQ feature transmitted to the decoder is encoded in $\log_2 B$ bits ($B$ is the number of items in the HQ codebook). To this end, we argue that the VQ process still underestimates the importance of the LQ feature for BIR. Directly alleviating this problem in the VQ process can potentially improve the performance of VQ-based BIR methods.

## 5 METHODOLOGY

### 5.1 REPLACING DISCRETE VQ SELECTION BY CONTINUOUS FEATURE TRANSFORMATION

Based on the three observations analyzed in §4, VQ is a two-sided coin with clear rewards and punishments for VQ-based BIR methods. To avoid the side effects of discrete VQ selection, motivated by the self-attention for code index prediction (Fig. 6 (a)), it is natural to employ cross-attention for continuous feature transformation from the LQ feature of input LQ image to HQ one with the HQ codebook. Specifically, as shown in Fig. 6 (b), to replace discrete VQ process, cross-attention takes the input feature as the query and the HQ codebook as both the key and value. The attention map correlates each LQ feature vector with HQ codebook items. Each output feature vector is an adaptively weighted combination of HQ codebook. In this way, the input LQ feature is transformed into HQ ones. However, the vanilla cross-attention ignores the self-expressiveness of LQ feature and would fail to preserve the fidelity of diverse LQ images (Figs. 1 (b) and (d)).

### 5.2 PROPOSED SELF-IN-CROSS-ATTENTION (SINCA)

To exploit useful self-expressiveness (Elhamifar & Vidal, 2013) of LQ images for better BIR performance, in this paper, we propose a new Self-in-Cross-Attention (SinCA) module to augment the HQ codebook with specific feature of input LQ image and performs cross-attention between input LQ feature and augmented codebook. As shown in Fig. 6 (c), given an LQ feature tensor

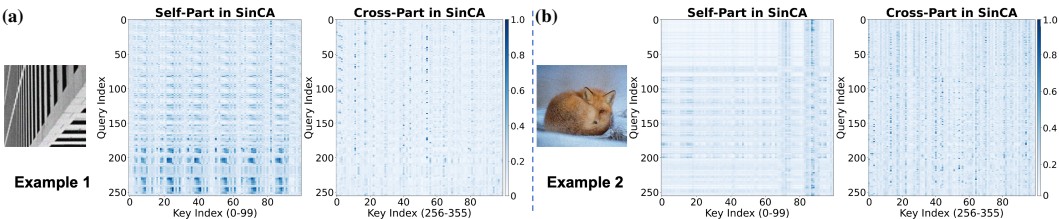

Figure 6: **(a) Self-attention** for code index prediction. **(b) Cross-attention** for feature transformation. **(c) Our proposed Self-in-Cross Attention (SinCA)** for effective feature transformation.

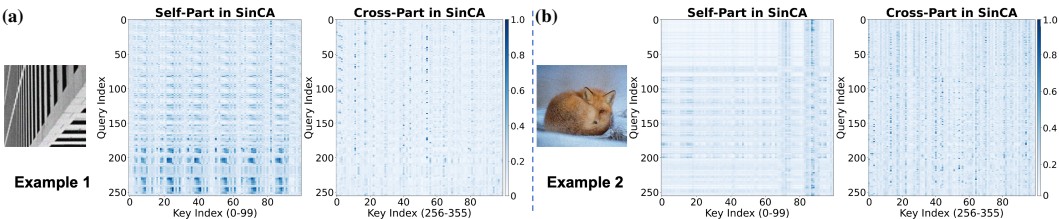

Figure 7: **Visualization of the attention map** of the first SinCA in 'FeMaSR+SinCA' for ×2 blind image super-resolution. We take inputs of size $64 \times 64$ which will be encoded into $16 \times 16 = 256$ feature vectors and visualize the attention weights by selecting the first 100 indices of **self-part** (0-99) and the first 100 indices of **cross-part** (256-355).

$\mathbf{z}^l \in \mathbb{R}^{h \times w \times d}$ of an LQ image extracted from the LQ encoder, our SinCA first reshapes it into an LQ feature matrix $\mathbf{X} \in \mathbb{R}^{hw \times d}$ and then multiplies it with a linear projection matrix $\mathbf{W_Q}$ to obtain the query matrix $\mathbf{Q}$. To jointly explore the expressiveness of HQ codebook $\mathbf{C} \in \mathbb{R}^{B \times d}$ and excavate the self-expressiveness of the LQ feature itself, we concatenate the LQ feature $\mathbf{X}$ and the HQ codebook $\mathbf{C}$ to obtain the key matrix $\mathbf{K} \in \mathbb{R}^{(hw+B) \times d}$ and the value matrix $\mathbf{V} \in \mathbb{R}^{(hw+B) \times d}$ with the corresponding linear projection matrices $\mathbf{W_K}$ and $\mathbf{W_V}$, respectively, as follows:

$$\mathbf{Q} = \mathbf{XW_Q}, \mathbf{K} = \begin{bmatrix} \mathbf{XW_K} \\ \mathbf{CW_K} \end{bmatrix}, \mathbf{V} = \begin{bmatrix} \mathbf{XW_V} \\ \mathbf{CW_V} \end{bmatrix}. \tag{3}$$

Denoting $\mathbf{K_X} = \mathbf{XW_K}$, $\mathbf{K_{Code}} = \mathbf{CW_K}$, $\mathbf{V_X} = \mathbf{XW_V}$, and $\mathbf{V_{Code}} = \mathbf{CW_V}$, the attention matrix $\mathbf{A} \in \mathbb{R}^{hw \times (hw+B)}$ and the output feature matrix $\mathbf{O}$ of our SinCA are computed as follows:

$$\mathbf{A} = \text{Softmax}\left(\frac{1}{\sqrt{d}}\mathbf{QK}^\top\right) = \text{Softmax}\left(\frac{1}{\sqrt{d}}\begin{bmatrix}\mathbf{QK_X^\top} & \mathbf{QK_{Code}^\top}\end{bmatrix}\right), \mathbf{O} = \mathbf{A}\begin{bmatrix}\mathbf{V_X} \\ \mathbf{V_{Code}}\end{bmatrix}. \tag{4}$$

As revealed by Eqn. (4), the attention map $\mathbf{A}$ correlates both the LQ feature and the input-augmented codebook. This is then used to adaptively weight the value feature matrices $\mathbf{V_X}$ and $\mathbf{V_{Code}}$. In this way, our SinCA simultaneously leverages the expressive HQ codebook prior and the self-expressiveness of input LQ image to obtain the output feature matrix $\mathbf{O}$.

We replace the VQ process in VQ-based BIR methods (Chen et al., 2022; Liu et al., 2023a; Zhou et al., 2022; TSAI et al., 2024) by a transformer (Dosovitskiy et al., 2021) using our SinCA, which aims to transform from the LQ feature of input image to HQ one with the HQ codebook.

### 5.3 A Closer Look at Our SinCA

To study the working mechanism of our SinCA, in Fig. 7 we visualize the attention map of "Fe-MaSR+SinCA" (variant of FeMaSR (Chen et al., 2022)) for ×2 blind image super-resolution. Here, the VQ in FeMaSR is replaced by a transformer using our SinCA. The "Example 1" in Fig. 7 (a) show highly structured with repeated patterns, which could be better represented by itself. Thus, the self-part in the attention map of our SinCA exhibits dense and high attention weights. This indicates that our SinCA effectively utilizes the self-expressiveness of the LQ feature itself for BIR. In contrast, the "Example 2" in Fig. 7 (b) presents an animal that can be well represented by the HQ codebook (cross-part). In this case, our SinCA utilizes more the HQ codebook to recover the HQ feature.

In sum, our SinCA utilizes augmented codebook to exploit the self-expressiveness of the LQ feature and the correlation between LQ feature and HQ codebook. This enables each pixel of input LQ image

to be adaptively recovered by a weighted combination of augmented codebook. Compared to discrete VQ process relying on HQ code index selection, our SinCA extends the representational range of the HQ codebook and further exploits the self-expressiveness of LQ image for VQ-based BIR methods.

# 6 EXPERIMENTS

## 6.1 EXPERIMENTAL SETUP

**Baselines**. We evaluate our Self-in-Cross-Attention (SinCA) on four typical VQ-based BIR methods: FeMaSR (Chen et al., 2022) and AdaCode (Liu et al., 2023a) for blind image super-resolution (BSR), CodeFormer (Zhou et al., 2022) and DAEFR (TSAI et al., 2024) for blind face restoration (BFR). For each baseline method, we directly use the HQ encoder, HQ codebook, and HQ decoder it learned in the first **Prior Learning** stage (§3) to reconstruct the HQ images. Then we replace the VQ process by a transformer with our SinCA, and fine-tuned the encoder to restore the LQ images along with fixed codebook and decoder learned by each baseline. For all models in our experiments, we set the number of attention heads as eight in the transformers using our SinCA and the channel dimension of both the input and the output of the transformer equal the channel dimension $d$ of codebook. More details will be provided in the Appendix.

**Training Dataset**. For BSR task, we train models on DIV2K (Agustsson & Timofte, 2017) training set, including 800 HQ images of 2K resolution. The HR images are cropped into $256 \times 256$ patches for training. Following the settings in FeMaSR and Adacode, we generate pairs of high-resolution (HR) and low-resolution (LR) training images with the degradation pipeline in BSRGAN (Zhang et al., 2021). For the BFR task, we employ the FFHQ (Karras et al., 2019) dataset with 70,000 HQ images of size $1024 \times 1024$. All HQ face images are resized into $512 \times 512$ for training. The LQ images are synthesized by following the degradation pipeline in CodeFormer or DAEFR, respectively.

**Test Set**. The BSR methods are evaluated on the DIV2K validation set, Urban100, BSDS100, and Manga109. The LR images are generated with the mixed degradation pipelines of (Zhang et al., 2021; Wang et al., 2021c) used in FeMaSR. The BFR methods are evaluated on the 3000 images used in (Zhou et al., 2022; TSAI et al., 2024) from CelebA-Test set (Karras et al., 2018) and the real-world dataset CelebChild-Test (Wang et al., 2021b). The LQ images from CelebA-Test set are synthesized with the same settings used in CodeFormer and DAEFR, respectively.

**Metrics**. For BSR, we report PSNR and SSIM results computed on the y-channel, and LPIPS (Zhang et al., 2018b) on RGB images. For BFR, we compute PSNR, SSIM, and LPIPS on CelebA-Test, while FID (Heusel et al., 2017) and NIQE (Mittal et al., 2012) on real-world CelebChild-Test.

**Training Details**. For each baseline using our SinCA, we follow its original setting of codebook size, optimizer, and learning rates. For FeMaSR and AdaCode, we train the second stage with a batchsize of 16 for 100K and 160K iterations, respectively. For CodeFormer, we train the stage-2 with a batchsize of 8 and stage-3 with a batchsize of 4. For DAEFR, we train the last stage with a batchsize of 8 for 200K iterations. The models are implemented by PyTorch. The BSR methods are trained with 2 RTX 4090 GPUs, while BFR methods are trained with 1 Tesla H100 GPU.

## 6.2 COMPARISON RESULTS

**Blind Image Super-Resolution**. In Table 1, we summarize the quantitative results of comparison methods on four benchmarks. One can see that the FeMaSR and AdaCode using our SinCA (denoted as "FeMaSR+SinCA" and "AdaCode+SinCA", respectively) outperform their corresponding baselines (retrained under the same settings) by 0.70~1.10dB in PSNR, 0.01~0.03 in SSIM, and generally better results in LPIPS. For reference, in Table 1 we also report the results of the released models, denoted as "FeMaSR (release)" and "AdaCode (release)". Note that albeit being trained with much larger datasets, the released models still achieve worse results than the corresponding methods using our SinCA in terms of PSNR and SSIM.

**Blind Face Restoration**. In Table 2, we provide the quantitative results of blind face restoration on synthetic and real-world datasets. One can see that, on CelebA-Test set, compared with retrained ones, CodeFormer using our SinCA obtains a gain of 0.23 dB and 0.0129 on PSNR and SSIM, respectively, while DAEFR using our SinCA achieves an improvement of 1.92 dB, 0.0636 and 0.0069 on PSNR,

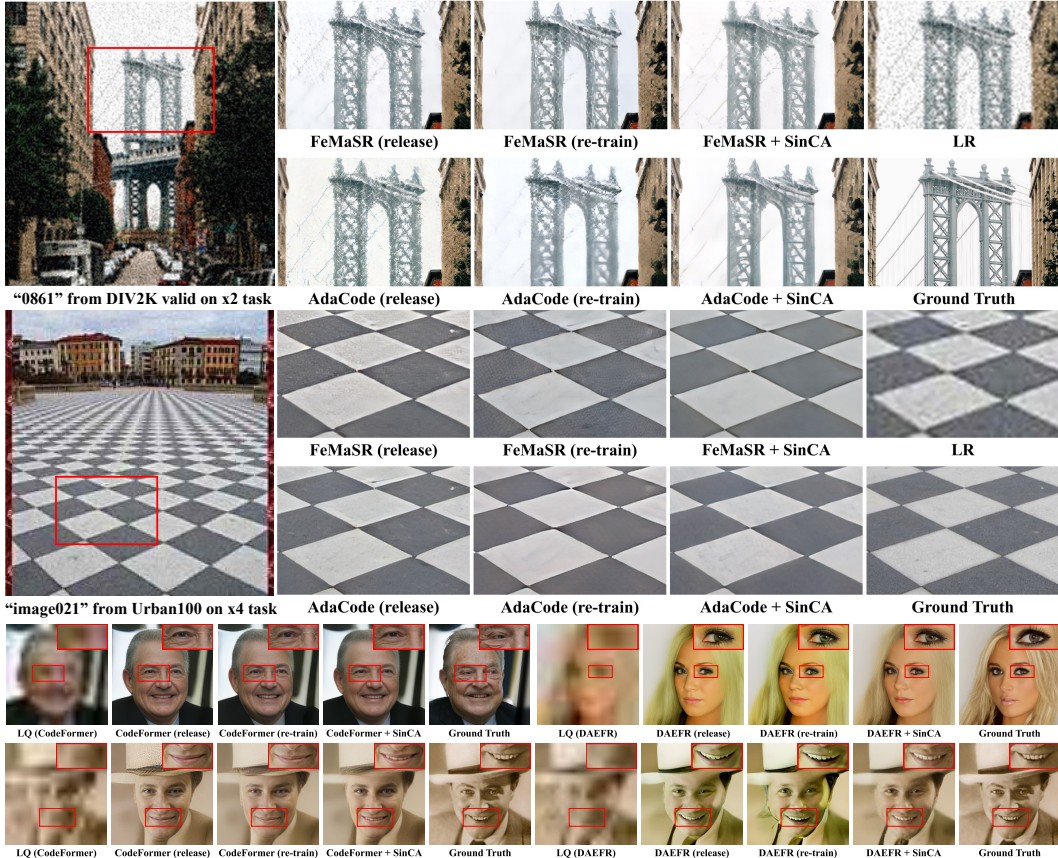

Figure 8: Comparison on blind image super-resolution and blind face restoration. "+SinCA": employing a transformer using our SinCA for continuous feature transformation.

Table 1: **Results of blind image super-resolution methods on four benchmark test sets**.

| Scale | Method | Urban100 | | | BSDS100 | | | Manga109 | | | DIV2K valid | | |
|---|---|---|---|---|---|---|---|---|---|---|---|---|---|
| | | PSNR↑ | SSIM↑ | LPIPS↓ | PSNR↑ | SSIM↑ | LPIPS↓ | PSNR↑ | SSIM↑ | LPIPS↓ | PSNR↑ | SSIM↑ | LPIPS↓ |
| ×2 | FeMaSR (release) | 20.11 | 0.5769 | 0.3847 | 21.90 | 0.5189 | 0.4225 | 22.14 | 0.7075 | 0.3358 | 23.44 | 0.6443 | 0.3878 |
| | FeMaSR (re-train) | 19.61 | 0.5607 | 0.4103 | 21.25 | 0.5052 | **0.4315** | 21.85 | 0.7092 | 0.3505 | 22.76 | 0.6311 | 0.4129 |
| | FeMaSR + SinCA | **20.59** | **0.5853** | 0.3958 | **22.43** | **0.5335** | 0.4387 | **22.61** | **0.7222** | 0.3428 | **23.84** | **0.6498** | 0.4026 |
| | AdaCode (release) | 20.46 | 0.5886 | 0.3808 | 22.03 | 0.5173 | 0.4199 | 22.35 | 0.7097 | 0.3226 | 23.38 | 0.6467 | 0.3759 |
| | AdaCode (re-train) | 19.47 | 0.5565 | 0.4124 | 21.28 | 0.5014 | 0.4422 | 21.81 | 0.7038 | 0.3522 | 22.55 | 0.6231 | 0.4102 |
| | AdaCode + SinCA | **20.46** | **0.5924** | **0.3940** | **22.44** | **0.5440** | **0.4293** | **22.74** | **0.7306** | **0.3326** | **23.75** | **0.6621** | **0.3917** |
| ×4 | FeMaSR (release) | 18.52 | 0.4891 | 0.4358 | 20.49 | 0.4528 | 0.4647 | 18.85 | 0.6107 | 0.3945 | 21.72 | 0.5626 | 0.4418 |
| | FeMaSR (re-train) | 18.41 | 0.4729 | 0.4759 | **20.82** | **0.4585** | 0.4950 | 18.86 | 0.5999 | 0.4269 | 21.72 | 0.5634 | 0.4715 |
| | FeMaSR + SinCA | **19.11** | **0.4887** | **0.4707** | 20.80 | 0.4477 | **0.4928** | **19.47** | **0.6168** | **0.4267** | **22.30** | **0.5703** | **0.4673** |
| | AdaCode (release) | 18.71 | 0.4875 | 0.4444 | 20.71 | 0.4495 | 0.4752 | 19.00 | 0.6067 | 0.3955 | 21.80 | 0.5638 | 0.4432 |
| | AdaCode (re-train) | 17.94 | 0.4644 | 0.4796 | 19.75 | 0.4237 | 0.5025 | 18.62 | 0.5936 | 0.5607 | 20.73 | 0.5404 | 0.4760 |
| | AdaCode + SinCA | **18.72** | **0.4780** | **0.4660** | **21.15** | **0.4617** | **0.4882** | **19.51** | **0.6093** | **0.4229** | **22.09** | **0.5658** | **0.4587** |

Table 2: **Results of blind face restoration methods on two synthetic and real-world test sets**.

| Method | CelebA-Test | | | CelebChild-Test | | Method | CelebA-Test | | | CelebChild-Test | |
|---|---|---|---|---|---|---|---|---|---|---|---|
| | PSNR↑ | SSIM↑ | LPIPS↓ | FID↓ | NIQE↓ | | PSNR↑ | SSIM↑ | LPIPS↓ | FID↓ | NIQE↓ |
| CodeFormer (release) | 22.19 | 0.5957 | 0.3152 | 116.23 | 4.983 | DAEFR (release) | 19.92 | 0.5534 | 0.3880 | 105.70 | 4.143 |
| CodeFormer (re-train) | 22.66 | 0.6248 | **0.3100** | 116.53 | **4.883** | DAEFR (re-train) | 19.65 | 0.5456 | 0.3675 | 105.23 | 4.220 |
| CodeFormer + SinCA | **22.89** | **0.6377** | 0.3108 | 121.50 | 5.112 | DAEFR + SinCA | **21.57** | **0.6092** | **0.3606** | **104.56** | **4.097** |

SSIM, LPIPS, respectively. On CelebChild-Test dataset, we cannot evaluate their performance on restoration fidelity since there is no ground-truth image. The CodeFormer using our SinCA obtain similar results of FID and NIQE to the released model, while DAEFR using our SinCA achieves an improvement in FID and NIQE. These results demonstrate that our SinCA serves as a promising replacement for the discrete VQ process for blind face restoration.

Table 3: **Comparison of using our SinCA for code index prediction or feature fusion**.

| Method | ×2 Blind Image Super-Resolution | | | | | | Method | Blind Face Restoration | | |
|---|---|---|---|---|---|---|---|---|---|---|
| | DIV2K valid | | | Urban100 | | | | CelebA-Test | | |
| | PSNR↑ | SSIM↑ | LPIPS↓ | PSNR↑ | SSIM↑ | LPIPS↓ | | PSNR↑ | SSIM↑ | LPIPS↓ |
| FeMaSR (index) | 22.85 | 0.6399 | 0.4124 | 19.78 | 0.5666 | 0.4101 | CodeFormer (index) | 22.63 | 0.6243 | **0.3061** |
| FeMaSR (feature) | **23.84** | **0.6498** | **0.4026** | **20.59** | **0.5853** | **0.3958** | CodeFormer (feature) | **22.89** | **0.6377** | 0.3108 |
| AdaCode (index) | 22.17 | 0.5542 | 0.4992 | 20.04 | 0.5635 | 0.4045 | DAEFR (index) | 19.85 | 0.5551 | 0.3635 |
| AdaCode (feature) | **23.75** | **0.6621** | **0.3917** | **20.46** | **0.5924** | **0.3940** | DAEFR (feature) | **21.57** | **0.6092** | **0.3606** |

Table 4: **Comparison of Self-Attention (SA), Cross-Attention (CA), and our Self-in-Cross-Attention (SinCA)** used by transformers for feature fusion in VQ-based BIR methods.

| Method | ×2 Blind Image Super-Resolution | | | | | | Method | Blind Face Restoration | | |
|---|---|---|---|---|---|---|---|---|---|---|
| | DIV2K valid | | | Urban100 | | | | CelebA-Test | | |
| | PSNR↑ | SSIM↑ | LPIPS↓ | PSNR↑ | SSIM↑ | LPIPS↓ | | PSNR↑ | SSIM↑ | LPIPS↓ |
| FeMaSR + SA | 23.01 | 0.6352 | 0.4035 | 19.87 | 0.5689 | 0.3990 | CodeFormer + SA | 22.85 | 0.6341 | 0.3165 |
| FeMaSR + CA | 23.22 | 0.6424 | **0.3970** | 20.08 | 0.5740 | **0.3957** | CodeFormer + CA | 22.83 | 0.6310 | 0.3154 |
| FeMaSR + SinCA | **23.84** | **0.6498** | 0.4026 | **20.59** | **0.5853** | 0.3958 | CodeFormer + SinCA | **22.89** | **0.6377** | **0.3108** |
| AdaCode + SA | 21.88 | 0.6053 | 0.4185 | 19.06 | 0.5379 | 0.4228 | DAEFR + SA | 21.53 | 0.5987 | 0.4118 |
| AdaCode + CA | 22.67 | 0.6349 | **0.3654** | 19.53 | 0.5583 | 0.4247 | DAEFR + CA | 21.47 | 0.6001 | 0.3770 |
| AdaCode + SinCA | **23.75** | **0.6621** | 0.3917 | **20.46** | **0.5924** | **0.3940** | DAEFR + SinCA | **21.57** | **0.6092** | **0.3606** |

**Visual Comparisons** in Fig. 8 show that, the four VQ-based BIR methods using our SinCA consistently preserves the colors and geometric shapes. For example, "FeMaSR + SinCA" and "AdaCode + SinCA" restore the steel ropes on the bridge and colors of bricks in the 1-st and 2-nd rows, respectively, while "CodeFormer + SinCA" and "DAEFR + SinCA" properly recover the skin tones and grin in the 3-rd and 4-th rows, respectively.

## 6.3 ABLATION STUDY

**Necessity of Replacing Index Prediction with Feature Transformation**. To study this aspect, we additionally design a variant of transformer using our SinCA for discrete index prediction in VQ-based BIR methods. As shown in Table 3, the four methods using our SinCA for feature transformation ("feature") obviously outperform those for code index prediction ("index"), especially on PSNR and SSIM. This validates the effectiveness of replacing the VQ process for code index prediction by feature transformation using our SinCA in VQ-based BIR methods.

**Effectiveness of our SinCA**. Here, we compare VQ-based BIR methods with transformers using self-attention (SA) (Dosovitskiy et al., 2021), cross-attention (CA) (Chen et al., 2021), or our SinCA for continuous feature learning. For CA, we generate the query/value matrix from HQ codebook and the key matrix from LQ feature for consistent input and output dimensions. As shown in Table 4, compared with those using SA and CA, the baselines using our SinCA achieve superior results on objective metrics in most cases. This validates the effectiveness of SinCA in VQ-based BIR methods.

## 7 CONCLUSION

In this paper, we revisited the key VQ process in VQ-based blind image restoration (BIR) methods and provided three close observations on the side-effects of VQ on code index prediction. We revealed that discrete VQ limits the representational capability of HQ codebook prior, is error-prone on code index prediction, and under-values the important LQ feature for BIR. Based on these observations, we proposed to replace the discrete VQ selection by continuous feature transformation from LQ feature to HQ ones via cross-attention of LQ feature and HQ codebook. We further proposed a Self-in-Cross-Attention (SinCA) module to augment HQ codebook with LQ feature and perform cross-attention between LQ feature and input-augmented codebook. Our SinCA extends the representational capability of HQ codebook and well leverages the self-expressiveness of input LQ image. Experiments demonstrated that, the four VQ-based BIR methods replacing the discrete VQ process with a transformer using our SinCA achieve better performance on blind image super-resolution and blind face restoration. Ablation studies also validated the effectiveness of our SinCA.

## 8 ETHICS STATEMENT

Our work adheres to ethical standards. We ensure that our work does not perpetuate bias or harm. All datasets used in our experiments are publicly available and ethically sourced, with proper consideration for privacy and consent.

## 9 REPRODUCIBILITY STATEMENT

We are committed to reproducibility. We provide the working mechanism of our proposed module in Sec. §5, our experimental setup in Sec. §6, and implementation details in Appendix A and Appendix B. All datasets used in our experiments are publicly available, and we will also release our code to facilitate further research and reproducibility.

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

## A  MORE ARCHITECTURE DETAILS

We develop transformers (Dosovitskiy et al., 2021) using our SinCA for feature transformation to replace the discrete VQ process for code index prediction in four VQ-based BIR methods. To verify the effectiveness of our SinCA, we apply the transformers using our SinCA on FeMaSR (Chen et al., 2022) and AdaCode (Liu et al., 2023a) for blind super-resolution as well as CodeFormer (Zhou et al., 2022) and DAEFR (TSAI et al., 2024) for blind face restoration. The transformers contain multiple layers. Each transformer layer incorporates one SinCA to excavate the useful information from input LQ feature itself and HQ codebook prior. Following our SinCA, each layer employs a Gated-Dconv Feed-Forward Network from (Zamir et al., 2022). We also use a skip connection to concatenate the input LQ feature of the transformer with the output HQ feature, followed by a linear layer for feature fusion. In all experiments, the input and output of the transformers maintain the same channel dimension as that of the HQ codebook in different baseline methods.

**FeMaSR** (Chen et al., 2022). The transformer using our SinCA for FeMaSR contains nine layers. Since the HQ codebook in FeMaSR is of size $1024 \times 512$, where the number of code items in codebook is $B = 1024$. The input and output channel dimension in our SinCA should be $d = 512$.

**AdaCode** (Liu et al., 2023a) has five categories of basis codebooks in its network backbone: architecture, indoor objects, natural scenes, street views and portraits, with codebooks of size $512 \times 256$, $256 \times 256$, $512 \times 256$, $256 \times 256$, and $256 \times 256$, respectively. We replace the VQ process for each category by a transformer using our SinCA with a channel dimension of $d = 256$, and $B = 512, 256, 512, 256, 256$. We set the number of transformer layers for these five categories to be 5, 4, 3, 4, and 4, respectively. The number of transformer layers is determined by the average value of weight maps from the pre-trained weight predictor.

**CodeFormer** (Zhou et al., 2022) uses a transformer (Dosovitskiy et al., 2021) consists of nine transformer layers with a channel dimension of $d = 512$, followed by a linear projection for code index prediction. To equip CodeFormer with our SinCA, we directly replace its transformer for code index prediction by our alternative transformer containing nine transformer layers with a channel dimension of $d = 256$, and the number of code items in HQ codebook is $B = 1024$.

**DAEFR** (TSAI et al., 2024) also employs a nine-layer transformer for code index prediction. Similar to our practice on CodeFormer (Zhou et al., 2022), we replace its transformer with the transformer containing nine layers with a channel dimension of $d = 256$, and the number of code items in HQ codebook is $B = 1024$.

## B  EXPERIMENTAL DETAILS

### B.1  MORE DETAILS ON MAIN EXPERIMENTS

In this section, we elaborate more details on the main experiments. All of our implementations are built upon publicly released codes of the baseline methods. For better clarity, we denote $\hat{\mathbf{z}}$ as the quantized HQ feature obtained by VQ or the output HQ feature by the transformer using our SinCA.

**FeMaSR** (Chen et al., 2022) adopts a two-stage training pipeline. We directly take the pre-trained codebook and decoder from the first stage. Then we replace the VQ process in FeMaSR with the transformer using our SinCA, and train the modified FeMaSR for the second stage.

**AdaCode** (Liu et al., 2023a) employs a three-stage training pipeline. The first two stages aim to obtain high-quality codebook and decoder prior as well as a weight predictor, while the last stage fine-tunes the encoder and weight predictor for image restoration. We replace the VQ process in AdaCode with the transformer using our SinCA, and train the modified AdaCode on the third stage. We take the original loss functions used in AdaCode for training.

**CodeFormer** (Zhou et al., 2022) utilizes a three-stage training pipeline. The first stage learns the HQ generative prior of codebook and decoder. The second stage learns a transformer to predict code indices. The third stage aims for feature fusion of LQ and HQ features in the decoder. We replace the VQ process in CodeFormer with the transformer using our SinCA, and train the modified CodeFormer on the second and third stages. The second training stage of vanilla CodeFormer has two loss functions: a cross-entropy loss $\mathcal{L}_{\text{code}}^{\text{token}}$ to supervise the training of transformer for index prediction

and an $\ell_2$ loss $\mathcal{L}_{\text{code}}^{\text{feat}'}$ to align the extracted LQ feature $\mathbf{z}^l$ and the quantized HQ feature $\hat{\mathbf{z}}$. The training objective function $\mathcal{L}_{\text{tf}}$ of the second stage can be written as follows:

$$\mathcal{L}_{\text{code}}^{\text{token}} = \sum_{i=0}^{hw-1} -s_i \log(\hat{s}_i), \quad \mathcal{L}_{\text{code}}^{\text{feat}'} = \left\| \mathbf{z}^l - \text{sg}(\hat{\mathbf{z}}) \right\|_2^2, \quad \mathcal{L}_{tf} = \lambda_{\text{token}} \mathcal{L}_{\text{code}}^{\text{token}} + \mathcal{L}_{\text{code}}^{\text{feat}'}, \quad (5)$$

where $s_i$ represents the $i$-th element of the GT indices (defined in §4.1 while $\hat{s}_i$ denotes the $i$-th element of indices predicted by the vanilla transformer in CodeFormer and $\lambda_{\text{token}}$ is a hyper-parameter used to balance the two loss functions. Since our transformer performs feature transformation instead of code index prediction, we replace the cross-entropy loss $\mathcal{L}_{\text{code}}^{\text{token}}$ with a feature matching loss $\mathcal{L}_{\text{matching}} = \left\| \mathbf{z}^t - \text{sg}(\hat{\mathbf{z}}) \right\|_2^2$, where $\mathbf{z}^t$ denotes the LQ feature $\mathbf{z}^l$ after transformation. Setting equal weight to $\mathcal{L}_{\text{matching}}$ and $\mathcal{L}_{\text{code}}^{\text{feat}'}$, our training objective function in the second stage is

$$\mathcal{L}_{tf} = \underbrace{\left\| \hat{\mathbf{z}} - \mathbf{z}^t \right\|_2^2}_{\mathcal{L}_{\text{matching}}} + \underbrace{\left\| \mathbf{z}^l - \text{sg}(\hat{\mathbf{z}}) \right\|_2^2}_{\mathcal{L}_{\text{code}}^{\text{feat}'}}. \quad (6)$$

The training of the third stage in CodeFormer inherits the loss functions used in the second stage, with additional image-level $\ell_1$ loss function and a perceptual loss (Johnson et al., 2016). Following this setting, we keep the loss functions adopted in training the second stage of the CodeFormer with the transformer using our SinCA with the $\ell_1$ loss and the perceptual loss as well.

**DAEFR** (TSAI et al., 2024) is trained in three stages. Similar to the second stage in Code-Former (Zhou et al., 2022), the third stage of DAEFR aims to align the LQ feature with HQ codebook, by using a cross-entropy loss function and an $\ell_2$ loss function of Eqn. (6) as the training objective. We replace the VQ process in DAEFR with the transformer using our SinCA for feature learning. We train the revised DAEFR in the third stage using our feature matching loss $\mathcal{L}_{\text{matching}}$ defined above and the loss function $\mathcal{L}_{\text{code}}^{\text{feat}'}$ defined in Eqn. (6), with equal weights.

## B.2 MORE DETAILS ON ABLATION STUDY

**Transformer Using Our SinCA for Code Index Prediction**. As described in §6.3 of our main paper, we also apply the transformer using our SinCA for code index prediction in VQ-based BIR methods. To this end, we use the linear layer following the transformer developed by our SinCA to project the output feature map of size $hw \times d$ into the size of $hw \times B$, with $B$ is the number of code items in HQ codebook. The output feature matrix of the linear layer is transformed into a probability matrix $\mathbf{P}$ via a softmax operation, where $p_{ij}$ ($i = 1, ..., hw$, $j = 1, ..., B$) represents the probability that the $i$-th LQ feature vector select the $j$-th code item of HQ codebook. Then we use top-1 selection for each LQ feature vector. For FeMaSR (Chen et al., 2022) and AdaCode (Liu et al., 2023a), besides their original loss functions, we introduce the cross-entropy loss $\mathcal{L}_{\text{code}}^{\text{token}}$ to supervise the learning of code index prediction. For CodeFormer (Zhou et al., 2022) and DAEFR (TSAI et al., 2024), since the code index prediction is incorporated in their original implementations, we directly use the loss functions used in the second **Restoration** stage of the corresponding VQ-based BIR methods.

**Self-Attention (SA) for Feature Transformation**. We replace our SinCA by SA (Dosovitskiy et al., 2021) for feature transformation in the transformer of VQ-based BIR methods. The experimental setups remain the same with those methods using our SinCA, with the exception that the SA module only takes LQ feature as the input to obtain query, key, and value. The input and output channel dimensions of SA in FeMaSR (Chen et al., 2022), AdaCode (Liu et al., 2023a), CodeFormer (Zhou et al., 2022), and DAEFR (TSAI et al., 2024) are set as 512, 256, 256, and 256, respectively.

**Cross-Attention (CA) for Feature Transformation**. Similar to the experiments on SA-based feature transformation mentioned above, we conduct experiments on CA (Chen et al., 2021) by replacing our SinCA in transformer with CA (Chen et al., 2021) for feature transformation. CA uses LQ feature to obtain the query and HQ codebook to obtain the key and the value. For FeMaSR (Chen et al., 2022), the input LQ feature and HQ codebook have 512 channels. For the transformers used in AdaCode (Liu et al., 2023a), CodeFormer (Zhou et al., 2022), and DAEFR (TSAI et al., 2024), both the input LQ feature and HQ codebook have 256 channels.

## C  MORE VISUAL COMPARISON RESULTS

Here, we provide more visual comparison results of different methods on blind image super-resolution (SR) in Figs. 9∼11 for ×2 SR task and in Figs. 12∼13 for ×4 SR task. More visual comparison results of blind face restoration are provided in Figs. 14∼16.

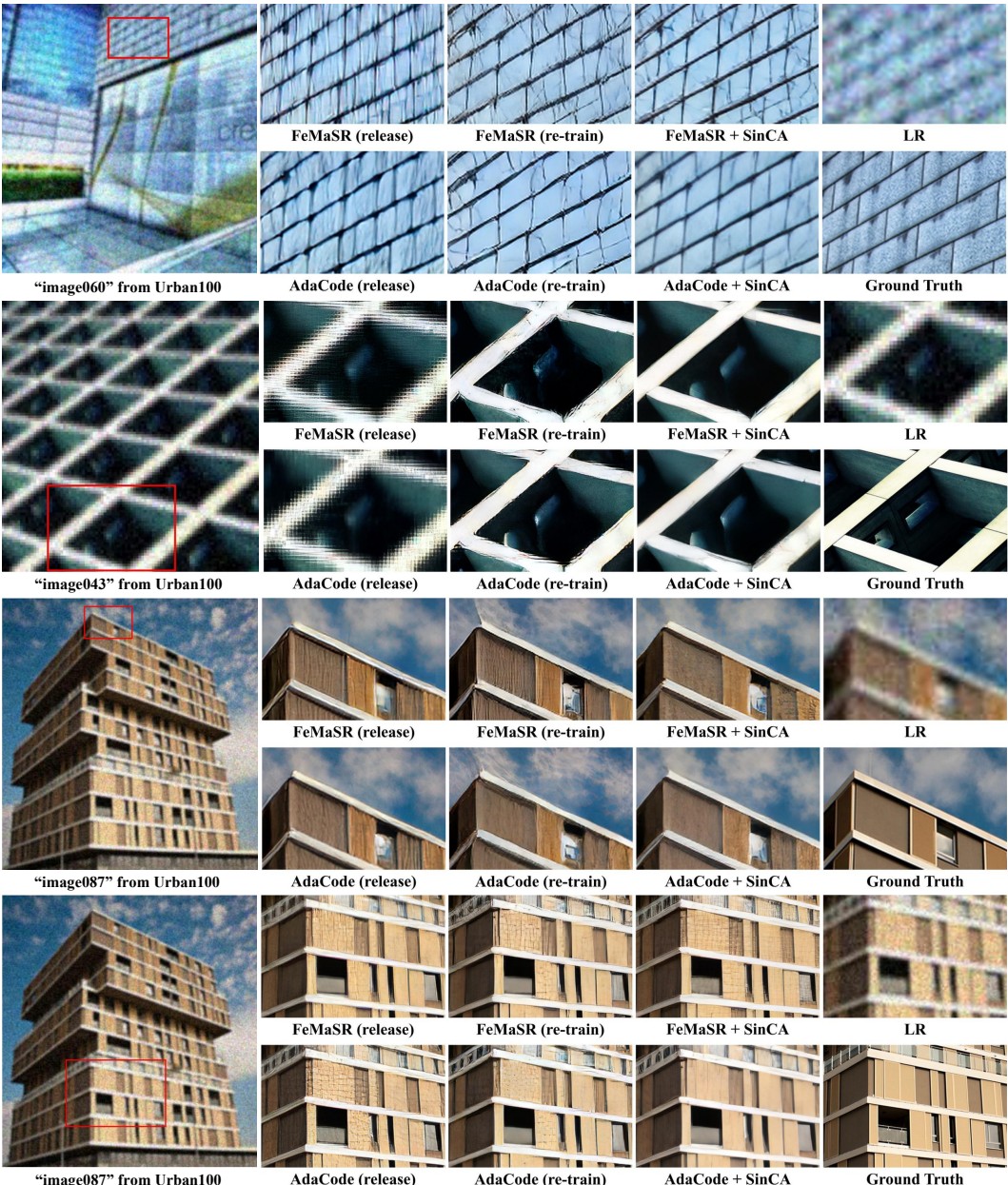

Figure 9: **More visual results on** ×2 **blind image super-resolution task.** "+SinCA": employing the transformer using our SinCA for feature transformation.

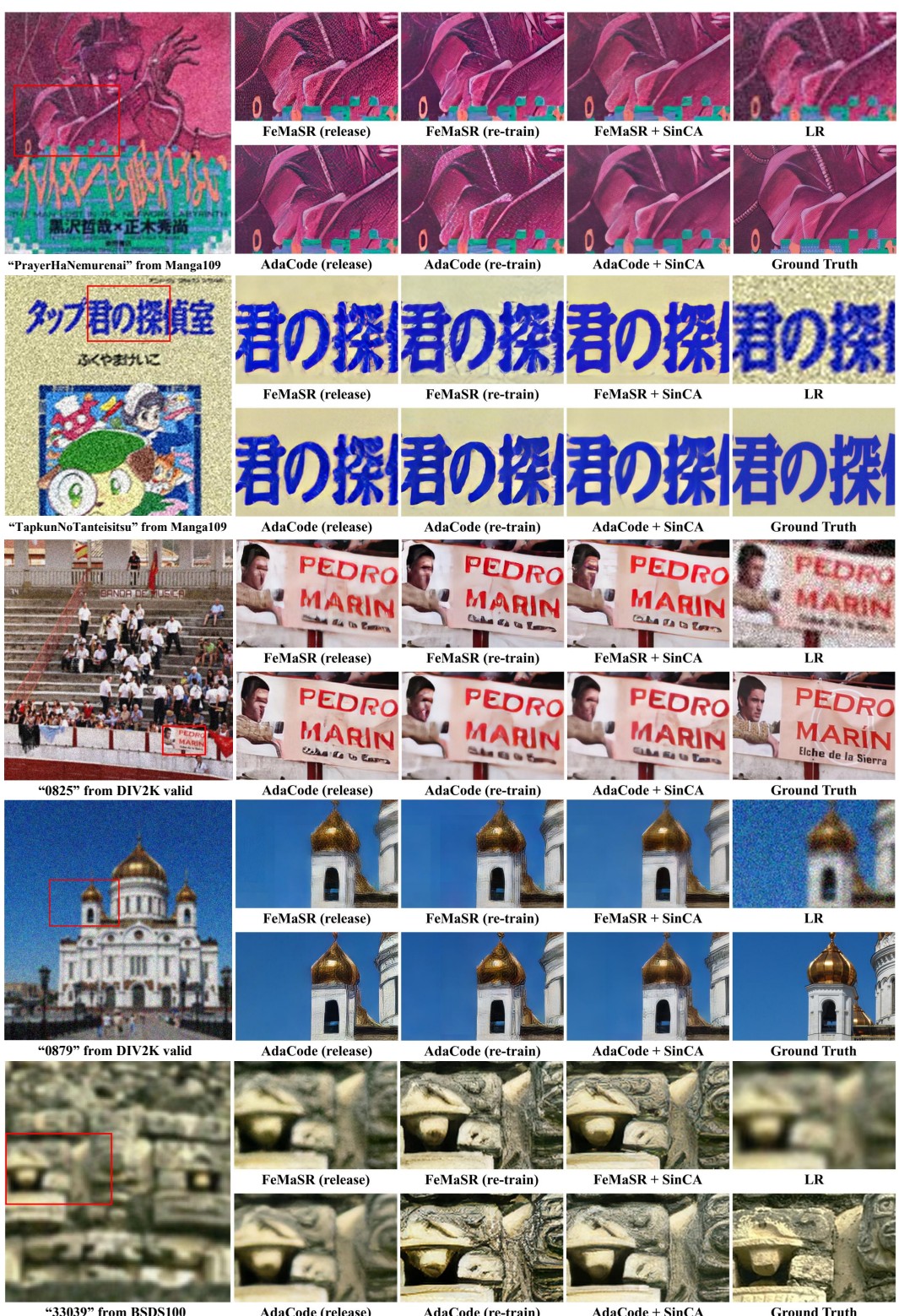

Figure 10: **More visual results on ×2 blind image super-resolution task.** "+SinCA": employing the transformer using our SinCA for feature transformation.

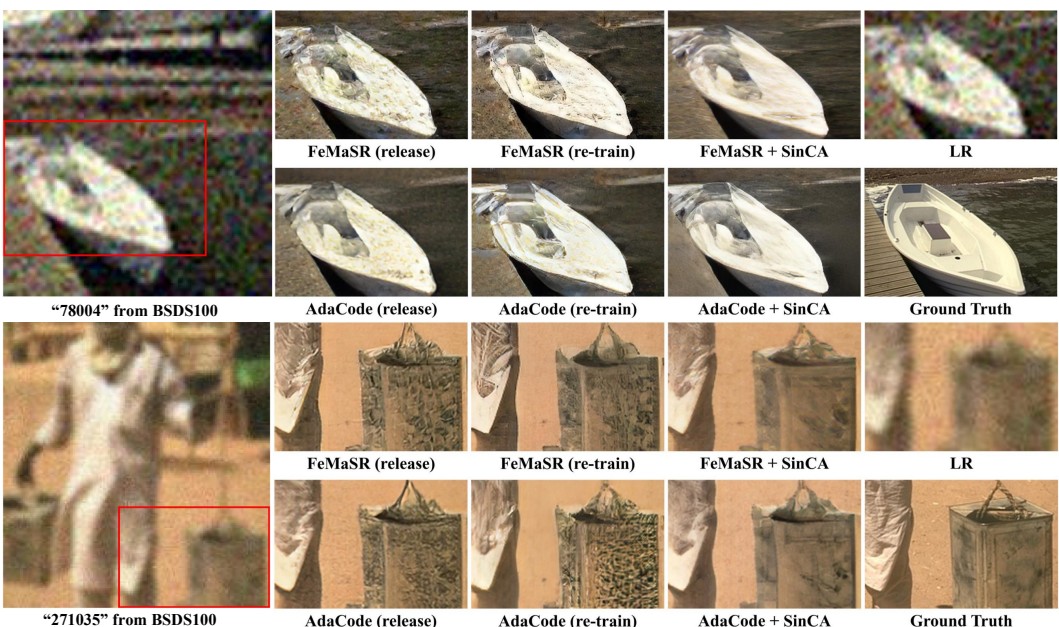

Figure 11: **More visual results on** $\times 2$ **blind image super-resolution task.** "+SinCA": employing the transformer using our SinCA for feature transformation.

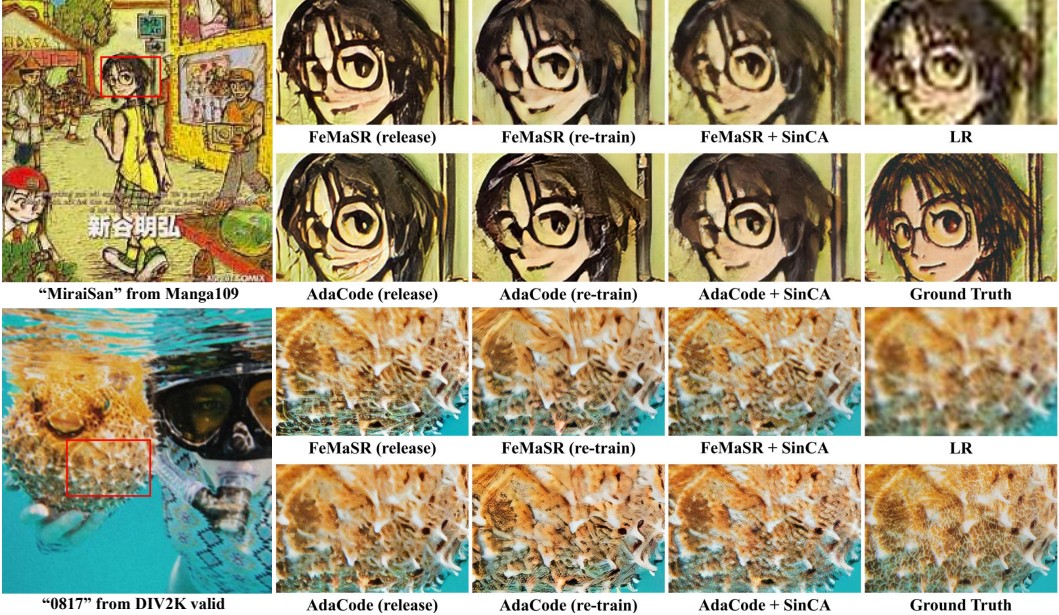

Figure 12: **More visual results on** $\times 4$ **blind image super-resolution task.** "+SinCA": employing the transformer using our SinCA for feature transformation.

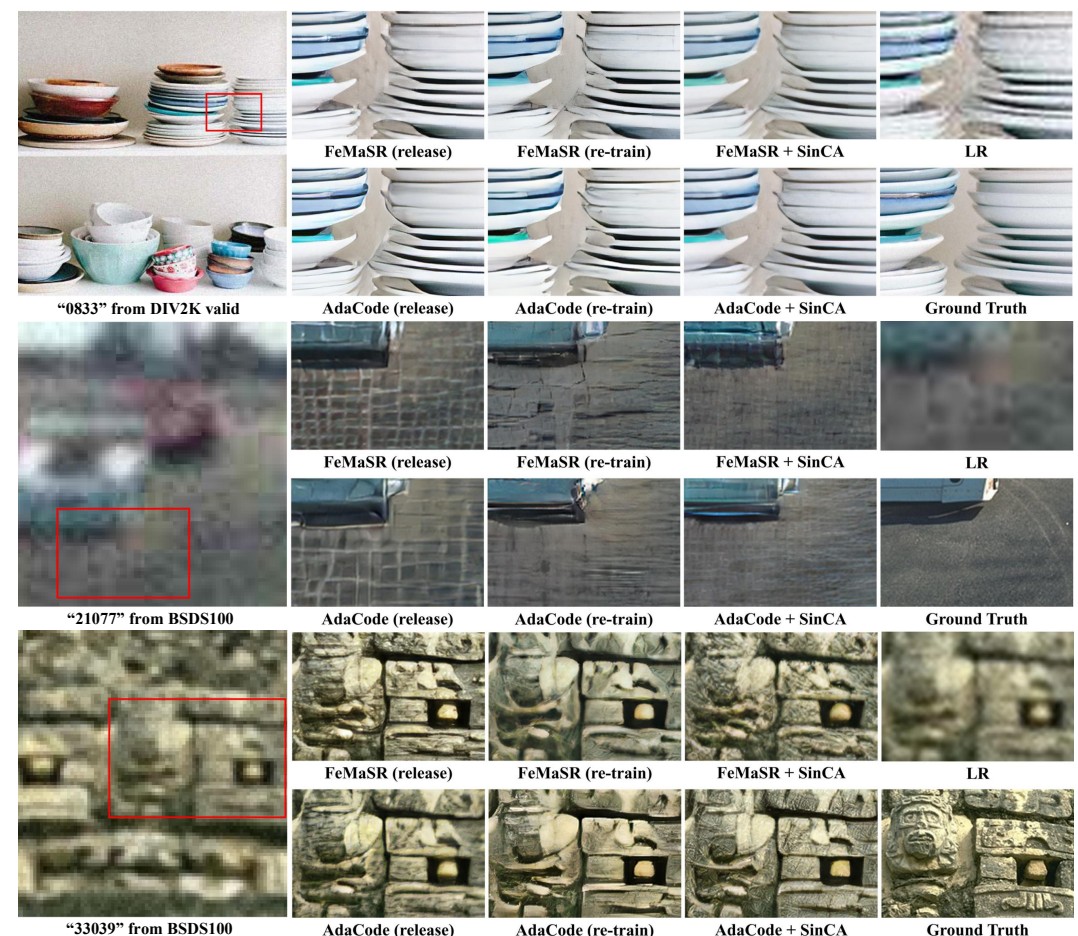

Figure 13: **More visual results on** $\times 4$ **blind image super-resolution task.** "+SinCA": employing the transformer using our SinCA for feature transformation.

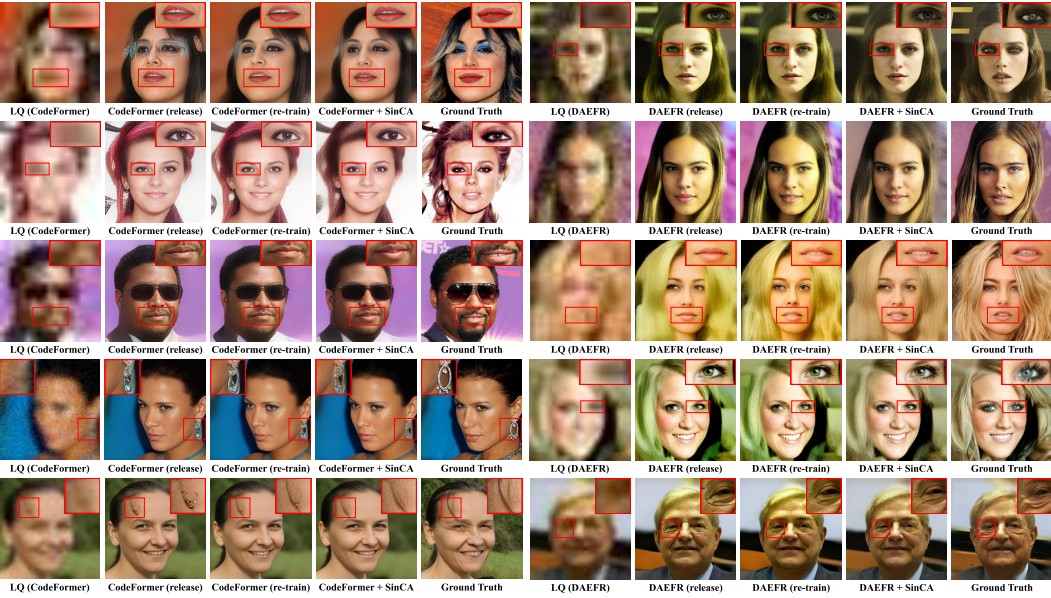

Figure 14: **More visual results of blind face restoration task on CelebA-Test.** "+SinCA": employing the transformer using our SinCA for feature transformation.

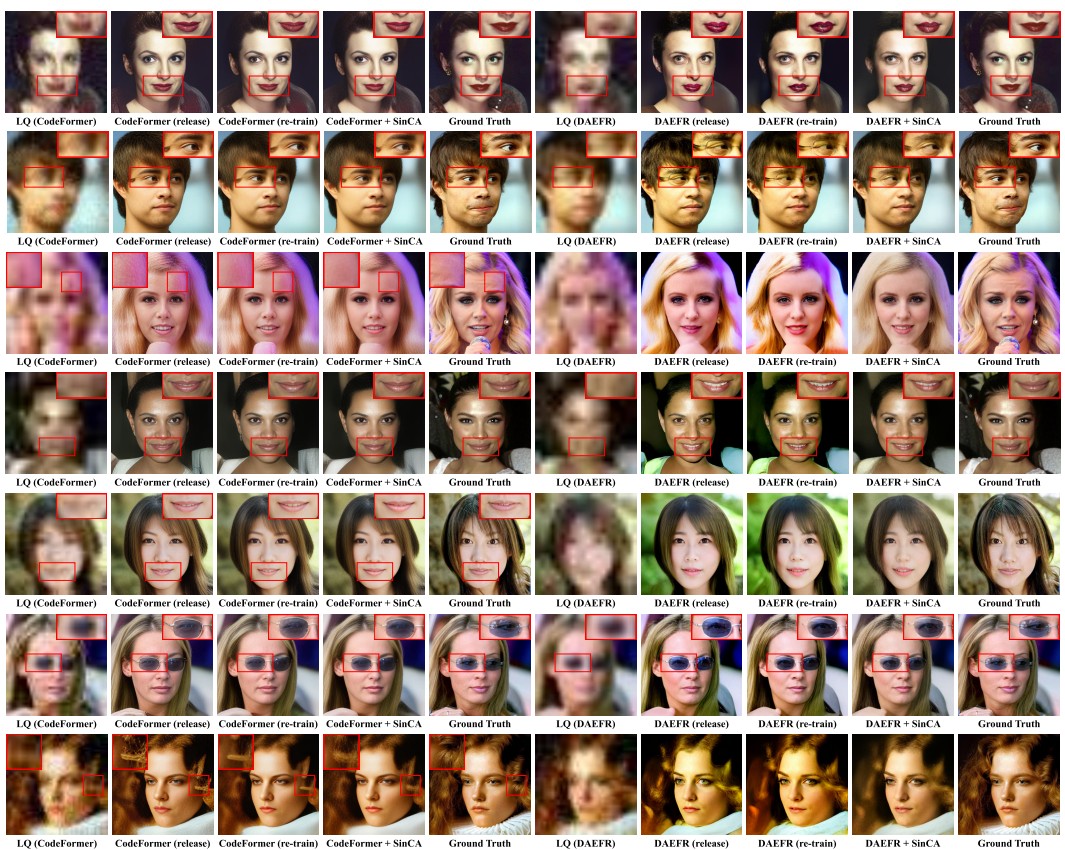

Figure 15: **More visual results of blind face restoration task on CelebA-Test.** "+SinCA": employing the transformer using our SinCA for feature transformation.

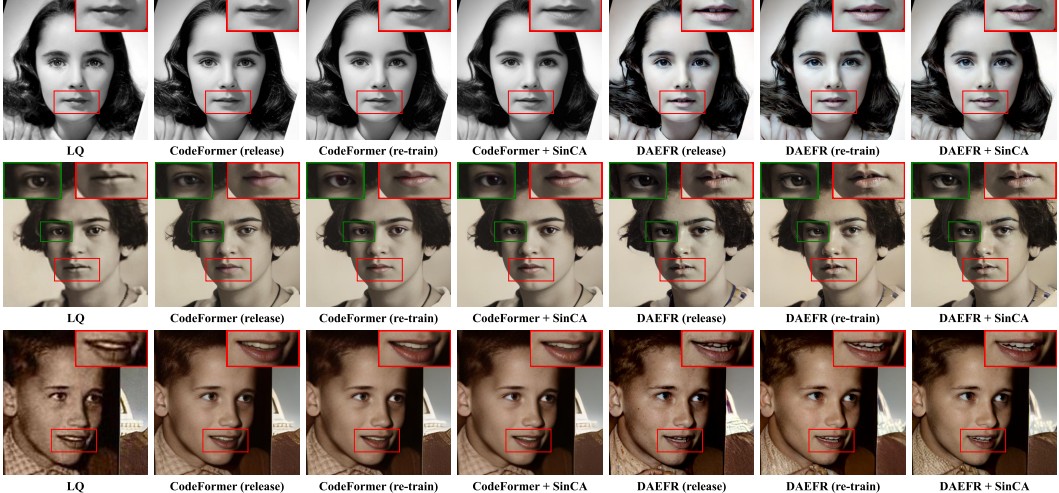

Figure 16: **More visual results of blind face restoration task on CelebChild-Test.** "+SinCA": employing the transformer using our SinCA for feature transformation.

