# OpenReview forum: "Revisiting Vector-Quantization for Blind Image Restoration"
_ICLR.cc/2025/Conference — Submitted to ICLR 2025_

### Official Review · Reviewer_r6SX · 2024-11-01

**Soundness:** 3
**Presentation:** 3
**Contribution:** 3
**Rating:** 6
**Confidence:** 4

**Summary:**

The paper explores the limitations of using Vector Quantization in blind image restoration tasks and highlights three key challenges: restricted representational capacity, prediction errors in code index selection, and undervaluing low-quality image features. To address these limitations, the authors propose a Self-in-Cross-Attention module, which augments the high-quality codebook by performing cross-attention between the HQ codebook and the LQ feature map. This module enables a continuous feature transformation process rather than relying on traditional discrete VQ, thereby enhancing the expressive power of HQ representations and improving restoration accuracy. Experimental results show that integrating SinCA into existing VQ-based BIR methods significantly improves performance on super-resolution and face restoration tasks.

**Strengths:**

+ It is interesting to see that the authors give a deep analysis of using Vector Quantization in blind image restoration tasks. The paper provides both quantitative metrics and visualizations to support the need for their proposed method.
+ The introduction of SinCA is a compelling solution to overcome VQ's limitations in blind restoration tasks by shifting from discrete selection to continuous feature transformation, expanding the capacity for HQ representation.
+ SinCA shows measurable improvement in image quality restoration, achieving higher metrics across various datasets for super-resolution and face restoration compared to traditional VQ-based methods.
+ The model is tested across multiple BIR benchmarks and configurations, demonstrating the broad applicability and effectiveness of the SinCA module in various restoration contexts.

**Weaknesses:**

- The additional complexity introduced by cross-attention mechanisms may lead to increased computational costs, which might limit its applicability in real-time or resource-constrained environments.
- The reliance on the HQ codebook for SinCA could lead to overfitting, especially if the training data for the HQ codebook does not fully represent the diversity of LQ images encountered in real-world settings.

**Questions:**

- In comparison with other methods, how does the introduction of the Self-in-Cross-Attention (SinCA) module affect computational complexity?
- In the quantitative comparison, the retrained methods have an obviously inferior performance than the released models. I wonder whether the authors did not re-train their models carefully.
- The degradation usually has some bias towards some specific degradation. In the visual comparison, it is better to provide more results on real-world LR images, rather than these used synthetic data.
- How does SinCA scale with larger or more complex codebooks, especially for high-resolution images? Does the performance or efficiency significantly change with codebook size, and how do the authors ensure the balance between restoration quality and computational load?

---

> ### Author Response · Authors · 2024-11-26
> **Response to Weakness 1 (W 1)  of Reviewer r6SX**
>
> Thank you for your detailed review and the helpful feedback to our work. We are grateful for your recognition of our **deep analysis of VQ-based BIR methods**, the **proposed SinCA as a compelling solution to overcome the limitations of VQ**, as well as the **broad applicability and effectiveness of our SinCA**.
>
> **W1: Increased computational costs.**
>
> As detailed in **Tab. 1**, we compare the computational costs and inference time of the traditional VQ process with our SinCA across different methods.
> It can be observed that SinCA generally consumes fewer computational resources compared to the transformers used for code index prediction in models like CodeFormer and DAEFR. This is because SinCA does not increase the dimension of the LQ feature, which is used for index prediction. While SinCA introduces additional computational costs compared to the negligible overhead of the nearest-neighbor feature matching employed in FeMaSR and AdaCode, this increase is justified by the substantial improvements in image restoration performance that SinCA provides.
>
> Furthermore, to address concerns regarding computational efficiency, we introduce four lightweight variants of SinCA: CodeFormer+SinCA (Light) with a single SinCA layer, DAEFR+SinCA (Light) with one SinCA layer, FeMaSR+SinCA (Light) with a single SinCA layer, and AdaCode+SinCA (Light) with five parallel SinCA layers, one for each of the five codebooks. We summarize the quantitative results in **Tabs. 2-3** and provide their computational costs and inference times in **Tab. 1**. These results demonstrate that even minimal implementations of SinCA can yield substantial performance improvements with minimal impact on computational resources. As a result, SinCA proves to be a viable and effective solution, even in resource-constrained scenarios.
>
>
> **Tab. 1  Computational cost summary**
> |Model|VQ Method|VQ FLOPs (G)|Total FLOPs (G)|Inference Time (ms)
> |-|-|-|-|-|
> |CodeFormer|transformers|10.02|809.72|32.87
> |CodeFormer+SinCA|SinCA| 5.59| 805.16|28.77
> |CodeFormer+SinCA (Light)|SinCA| 0.61| 800.18|27.06
> |DAEFR|transformers|10.02|908.30|65.58
> |DAEFR+SinCA|SinCA|5.59|903.74|64.95
> |DAEFR+SinCA (Light)|SinCA|0.61|898.76|54.60
> |FeMaSR|nearest-neighbor|2.15|212.27|75.73
> |FeMaSR+SinCA|SinCA|59.66|269.78|116.70
> |FeMaSR+SinCA（Light)| SinCA|7.59| 217.71| 73.09
> |AdaCode|nearest-neighbor|1.88|259.62|69.87
> |AdaCode+SinCA|SinCA |30.10|287.84|83.33
> |AdaCode+SinCA (Light)|SinCA |8.04|265.78|69.40
>
> For CodeFormer and DAEFR, FLOPs is calculated by input of size $512\times 512$, while for FeMaSR and AdaCode, FLOPs is obtained  by input of size $128\times 128$ on $\times2$ task. The inference time is calculated as the average over 100 forward passes.
>
> **Tab. 2  Results of blind image super-resolution**
> | Method|Urban100|||DIV2K|||B100|||Manga109|||
> |-|-|-|-|-|-|-|-|-|-|-|-|-|
> ||PSNR↑|SSIM↑|LPIPS↓|PSNR↑|SSIM↑|LPIPS↓|PSNR↑|SSIM↑|LPIPS↓|PSNR↑|SSIM↑|LPIPS↓
> |FeMaSR|19.39 |0.5438| 0.3987|22.64|0.6161|0.4039| 21.25|0.4314|0.4938|21.85|0.6770|0.3389
> |FeMaSR + SinCA (Light) |20.25 |0.5733| 0.3982|23.52|0.6447|0.3992|22.14|0.5238|0.4318|22.47|0.7193|0.3429
> |AdaCode|19.47|0.5565|0.4124|22.55|0.6231|0.4102|21.28|0.5014|0.4422|21.81|0.7038|0.3522
> |AdaCode + SinCA (Light)|20.23|0.5798|0.3945|23.34|0.6470|0.3957|21.62|0.5145|0.4350|22.33|0.7192|0.3359
>
> **Tab. 3 Results of blind face restoration**
> | Method|CelebA|||
> |-|-|-|-|
> || PSNR↑|SSIM↑|LPIPS↓
> |CodeFormer|22.66|0.6248|0.3100
> |CodeFormer+SinCA (Light)| 22.86|0.6380|0.3157
> |DAEFR|19.65|0.5456|0.3675
> |DAEFR+SinCA (Light)| 21.54|0.6096|0.3666

---

> > ### Author Response · Authors · 2024-11-26
> > **Response to Weakness 2 (W 2) and Question 1 (Q 1)  and Question 2 (Q 2) of Reviewer r6SX**
> >
> > **W2: Concern of overfitting**.
> >
> > We acknowledge the concern regarding the potential for overfitting due to the reliance on HQ codebook, especially when the training data may not fully represent the diversity of LQ images encountered in real-world scenarios. However, it is important to note that traditional VQ-based methods rely entirely on the HQ codebook since they directly replace LQ features with selected code items from HQ codebook. This approach makes them more prone to overfitting when faced with diverse or unseen LQ images.
> >
> > In contrast, as introduced in Sec. 5 in our main paper, SinCA introduces an augmented codebook in the cross-attention mechanism, which not only exploits the self-expressiveness of LQ features but also leverages the correlation between LQ feature and HQ codebook. This approach allows SinCA to adaptively balance the use of the HQ codebook with information derived from LQ features. By integrating LQ information into the feature transformation process, SinCA effectively expands the representation capacity of the HQ codebook and enhances the ability to handle diverse LQ images. As shown in Tabs. 1-2 and Fig. 8 in our main paper, this fundamental enhancement makes SinCA more robust and flexible, effectively mitigating the risk of overfitting while maintaining strong restoration performance in varied real-world conditions. We also provide more visual comparison in appendix and the results on real-world datasets in **Tab. 3** and **Figs. 6-12** of the attached PDF file (File name:'Visual\_Resulst\_2.pdf' and 'Visual\_Resulst\_3.pdf') in our **[anonymous link](https://anonymous.4open.science/r/ICLR2025-FC45/)** to validate the effectiveness of our SinCA under diverse scenarios.
> >
> > ---
> > **Q1: Effect on computational complexity**.
> >
> > The computational complexity of our SinCA is $O(hw\times d \times (hw+B))$, where $h$, $w$, and $d$ are the height, width and channel dimension of the LQ feature and $B$ is the number of codes in HQ codebook. Detailed computational costs and inference times are provided in **Tab. 1** . To further alleviate concerns about resource consumption, we develop lightweight variants of SinCA that reduce computational demands while still delivering substantial restoration performance improvements.
> >
> > ---
> > **Q2: Inferior performance on retrained methods**.
> >
> > We want to clarify that the training settings for the models compared in our study are maintained fairly. For CodeFormer and DAEFR, we utilized their entire original training dataset, which includes 70,000 images from the FFHQ dataset. For FeMaSR and AdaCode, since part of their original training set are randomly sampled from FFHQ and downsampled with random scale factor, we use the DIV2K dataset for training. It is important to note that the training process for our method and the retrained models follows exactly the same settings, ensuring a fair comparison across all evaluations.
> >
> > Furthermore, to provide a comprehensive perspective, we includes results from the originally released models in the quantitative comparison. As shown in Tab. 1 and Tab. 2 of our main paper, our proposed SinCA not only surpasses the retrained models under the same training settings, but also exhibits superior performance compared to the released models, which benefit from a larger training dataset.
> > This approach ensures a thorough and equitable evaluation of all methods being studied.
> >
> > ---
> > **Tab. 3  Results of blind image super-resolution on real-world datasets**
> > |Method|RealSRx2|||DRealSRx2|||RealSRx4|||DRealSRx4|||
> > |-|-|-|-|-|-|-|-|-|-|-|-|-|
> > ||PSNR↑|SSIM↑|LPIPS↓|PSNR↑|SSIM↑|LPIPS↓|PSNR↑|SSIM↑|LPIPS↓|PSNR↑|SSIM↑|LPIPS↓
> > |FeMaSR|25.10|0.7858|0.3380|25.92|0.7991|0.3397|23.49|0.6921|0.4356|25.31|0.7450|0.4312
> > |FeMaSR+SinCA|27.85|0.8178|0.3304|28.36|0.8295|0.3222|24.98|0.7094|0.4242|26.76|0.7635|0.4121
> > |AdaCode|25.01|0.7733|0.3420|26.08|0.7898|0.3373|22.78|0.6762|0.4274|24.63|0.7333|0.4206
> > |AdaCode+SinCA|27.13|0.8256|0.3133|27.91|0.8361|0.3098|24.31|0.6982|0.4164|26.08|0.7473|0.4121

---

> > > ### Author Response · Authors · 2024-11-26
> > > **Response to Question 3 (Q 3)  and Question 4 (Q 4) of Reviewer r6SX**
> > >
> > > **Q3: More experiments on real-world images**.
> > >
> > > We conduct more experiments on the real-world dataset to evaluate the proposed SinCA under various degradations on both BSR and BFR tasks.
> > > We present the visual comparison of BSR task in **Figs. 9-12** of the attached PDF file (File name:'Visual\_Resulst\_3.pdf') in our **[anonymous link](https://anonymous.4open.science/r/ICLR2025-FC45/)** on the RealSR and DrealSR datasets. The visual comparison of BFR are provided in **Figs. 6-8** of the attached PDF file (File name:'Visual\_Resulst\_2.pdf') in our **[anonymous link](https://anonymous.4open.science/r/ICLR2025-FC45/)** on WebPhoto-Test, WIDER-Test and LFW-Test. Clicking “view raw” can provide a better view.
> > > In addition, we also provide the quantative results of FeMaSR and AdaCode on real-world datasets in **Tab. 3** for further comparison.
> > > These results demonstrate that the proposed SinCA not only successfully restores fine details but also preserves the quality of the images, thereby confirming the effectiveness of our approach.
> > >
> > > ---
> > > **Q4: Concern of efficiency with larger codebook size and high-resolution**
> > >
> > > We explore the computational efficiency with following two aspects:
> > >
> > > **Larger or more complex codebook.**
> > > In our work, we conduct experiments on AdaCode to verify the effectiveness of SinCA. As detailed in our appendix (Page 15, Line 772), AdaCode incorporates five categories of basis codebooks in its network backbone: architecture, indoor objects, natural scenes, street views, and portraits, with codebook sizes of $512\times256$, $256\times256$, $512\times256$, $256\times256$, and $256\times256$, respectively. Even with these complex codebooks, as shown in **Tabs. 1-2**, our AdaCode+SinCA (Light) with one SinCA layer per codebook achieves performance improvements while maintaining computational efficiency.
> > >
> > > We also explore the impact of codebook size on DAEFR+SinCA (Light), which initially uses a $1024\times256$ codebook. We train three variants: one with 512 codebook items (using the first half of DAEFR's codebook), one with 1024 items, and one with 1536 items (combining DAEFR's codebook with part of CodeFormer’s due to limited time). As demonstrated, DAEFR+SinCA (Light) remains computationally efficient even with increasing codebook size, as summarized in **Tab. 4** .
> > >
> > > **High-resolution images.** For high-resolution images, blind image super-resolution methods often use tiling to divide the image into smaller patches for efficiency. Therefore we provide the computation cost per patch of size $128\times128$. As shown in **Tab. 1**, SinCA (Light) remains computationally efficient in these scenarios. However, for blind face restoration, where images exhibit strong structural features, tiling is not employed. We compare the computational cost of the VQ process for high-resolution face images ($1024\times1024$ and $512\times512$) using transformers in CodeFormer and DAEFR, as well as our SinCA. As shown in **Tab. 5**, SinCA is more efficient for high-resolution face restoration because it doesn't increase the dimension of LQ features used in transformers of CodeFormer and DAEFR for index prediction.
> > >
> > > In conclusion, our extensive evaluations demonstrate that SinCA effectively balances computational efficiency with enhancements in image restoration quality and fidelity.
> > >
> > > ---
> > > **Tab. 4 Results of DAEFR+SinCA (Light) with different codebook size**
> > > | Method|VQ FLOPs (G)|Total FLOPs (G)|Inference Time (ms) |CelebA|||
> > > |-|-|-|-|-|-|-|
> > > ||||| PSNR↑|SSIM↑|LPIPS↓
> > > |DAEFR+SinCA (Light), 512 code| 0.4778|898.49|54.54|21.45|0.6053|0.3772
> > > |DAEFR+SinCA (Light), 1024 code| 0.6137|898.76|54.60|21.54|0.6096|0.3666
> > > |DAEFR+SinCA (Light), 1536 code| 0.7497| 899.03 |55.28|21.54|0.6061|0.3690
> > >
> > > **Tab. 5 Computation cost for blind face restoration in VQ process**
> > > |Resolution|Method|VQ FLOPs (G)|
> > > |-|-|-|
> > > |512|using transformer|10.02
> > > |512| using SinCA|5.59
> > > |1024|using transformer|39.86
> > > |1024| using SinCA|15.14
> > >
> > > ---
> > > Once again, we sincerely appreciate your affirmation of our work with the constructive feedback and insightful suggestions you provided, which have been instrumental in enhancing our study.

---

> > > > ### Comment · Reviewer_r6SX · 2024-11-27
> > > > **Feedback for Response**
> > > >
> > > > Thanks to the authors for providing such a detailed response. My concerns about real-world effects, and model efficiency are well addressed. Normally, I should raise the score, but the insight of this work brought to this community (refer to other reviews) seems to be not enough to reach 8. I will keep this score temporarily and will carefully refer to other reviewers' concerns.

---

> > > > > ### Author Response · Authors · 2024-11-29
> > > > > **Response to Reviewer r6SX**
> > > > >
> > > > > Thanks for your positive feedback and your approval of our work. We have updated our reply to other reviewers, providing further clarification. With detailed observations, and significant improvements in model performance, we believe that our work provides meaningful insights for VQ-based blind image restoration. We sincerely hope to gain your continued support.

---

### Official Review · Reviewer_WGGn · 2024-11-02

**Soundness:** 3
**Presentation:** 3
**Contribution:** 3
**Rating:** 8
**Confidence:** 4

**Summary:**

In this article, the authors rethink the key VQ process in VQ-based BIR methods and, based on observations, raised three issues: 1. VQ limits the representational capacity of HQ code books; 2. VQ is prone to errors in code index prediction; 3. VQ underestimates the importance of low-quality features in BIR. To address these issues, the author proposed using continuous features to replace discrete VQ selection and introduced a novel Self-Cross Attention module (SinCA). This module enhances the features of HQ code books and LQ images and performs cross-attention between LQ features and the enhanced code books. The authors conduct extensive experiments on various datasets to demonstrate the effectiveness of the SinCA module in different blind super-resolution tasks, showing that this method can achieve better quantitative metrics and visual performance.

**Strengths:**

Advantages: 1.This paper proposes a novel Self-Cross Attention module and the use of continuous features to replace discrete VQ selection, providing a different perspective for subsequent blind super-resolution research. 2.The observations made in this paper are quite reasonable and the arguments are well-substantiated. 3.The writing of this paper is fluent and generally conforms to academic writing standards.

**Weaknesses:**

On page 4, lines 174-178, the paper introduces the overall loss function, which is composed of reconstruction loss, code book loss, and commitment loss. However, the paper seems to lack a study of the loss function. If the paper could supplement this aspect, it would make the argument more complete.
On page 5, in Figure 3, the connection between the images and the text appears not to be very tight. If this issue could be improved, it would make the logical argumentation of the paper more fluid.
In the experimental part of the paper, the author conducted numerous experiments on different datasets. However, since it is in the field of blind super-resolution, if experiments on image recovery under adverse conditions could be added, it would make the experiments in the paper more comprehensive.
Conclusion: If the aforementioned shortcomings could be addressed, this article should be accepted.

**Questions:**

On page 4, lines 174-178, the paper introduces the overall loss function, which is composed of reconstruction loss, code book loss, and commitment loss. However, the paper seems to lack a study of the loss function. If the paper could supplement this aspect, it would make the argument more complete.
On page 5, in Figure 3, the connection between the images and the text appears not to be very tight. If this issue could be improved, it would make the logical argumentation of the paper more fluid.
In the experimental part of the paper, the author conducted numerous experiments on different datasets. However, since it is in the field of blind super-resolution, if experiments on image recovery under adverse conditions could be added, it would make the experiments in the paper more comprehensive.
Conclusion: If the aforementioned shortcomings could be addressed, this article should be accepted.

---

> ### Author Response · Authors · 2024-11-26
> **Response of Reviewer WGGn**
>
> Thanks for your thorough review and the valuable feedback on our work. We sincerely appreciate your recognition of the **novelty of the proposed SinCA**, the **reasonableness and sufficiency of our observations**, as well as the **presentation of our manuscript**.
>
> **W1: More study of reconstruction loss**.
>
> In Sec. 3, page 4 of our main paper, we describe the total loss function employed in VQVAE and VQGAN for image generation. As the VQ-based BIR methods integrate the VQ process into the restoration framework, they incorporate modifications to the loss function to better align with restoration objectives. Since we mainly focus on replacing the discrete vector-quantization process with a more effective alternative, we maintain the original training strategy and follow the loss functions employed in these VQ-based BIR methods. Specifically, for CodeFormer and DAEFR, which employ cross-entropy loss to supervise transformer learning for index prediction, we substitute cross-entropy loss with a feature matching loss to accommodate the continuous learning of our SinCA, as detailed in our appendix (Page 16, line 821). For our feature matching loss, we adhere to the common implementation of code-level losses, using L2 loss.
>
> Inspired by your feedback, we exploring the use of L1 loss as a feature matching loss to supervise the learning of the SinCA on DAEFR. As shown in **Tab. 1**, L1 loss could potentially enhance the restoration performance on synthetic CelebA-Test. We recognize the potential of further exploring the design of code-level loss and consider it as a promising future direction. Thanks for the constructive comment.
>
> ---
>
> **W2: Clarify Fig. 3**.
>
> We have revised the illustration in Fig. 3 of our main paper to ensure a tighter connection between the images and the text as follows.
>
> **T-SNE visualization of VQ process** in FeMaSR (a) and CodeFormer (b). Different code items in HQ codebook are marked by "☆" in different colors. The color of  LQ feature vector marked by "◯'' or  the HQ feature vector marked by "△'' is the same with the codebook item they select in VQ process. Gray dashed lines "- -'' connects the LQ feature vector "◯'' and its selected codebook item "☆"  (“VQ Process”). Red dash lines "- -'' connects the LQ feature vector  "◯''  and the codebook item "☆" selected by the corresponding HQ feature vector using nearest-neighbor (NN) feature matching in the **Prior Learning** Stage (“GT Selection”).
> (a) NN feature matching on LQ feature vectors are inconsistent with “GT Selection”. For a given LQ feature vector, the codebook item selected by NN feature matching (gray dash line, "- -'') is quite different from the corresponding "GT Selection'' (red dashed line, "- -'').
>  (b) Transformer for code index prediction is not robust to image degradation. In a degraded LQ image, an "LQ Eye'' patch looks like skin area and selects the code item represented by many "HQ Skin'' patches. The regions marked by purple dashed box and orange dashed box are enlarged for better view ("Enlarge'').
>  (c) Prediction accuracy of code indices by transformer and SSIM results achieved by CodeFormer using "LQ Indices'' predicted on LQ images, "HQ Indices'' predicted on HQ images, or "GT Indices'' defined in Sec.4.2.
>
> ----
>
> **W3: More experiments on adverse conditions for BSR**.
>
> We conduct more experiments on the real-world dataset to evaluate the proposed SinCA under adverse conditions on both $\times 2$ and $\times 4$ task. We summarize the quantitative results in **Tab. 2**, verifying that our SinCA can achieve superior performance on  adverse senarios. The qualitative results are provided in **Figs. 9-12** of the attached PDF file (File name:'Visual\_Resulst\_3.pdf') in our **[anonymous link](https://anonymous.4open.science/r/ICLR2025-FC45/)**, which also demonstrate the effectiveness of our SinCA. Clicking “view raw” can provide a better view.
>
> ----
>
> **Tab. 1 Implementation of feature matching loss on DAEFR**
> | Method|CelebA|||
> |-|-|-|-|
> || PSNR↑|SSIM↑|LPIPS↓
> |L2 loss|21.57|0.6092|0.3606
> |L1 loss|21.68|0.6100|0.3556
>
> **Tab. 2 Results of BSR on real-world datasets**
> |Method|RealSRx2|||DRealSRx2|||RealSRx4|||DRealSRx4|||
> |-|-|-|-|-|-|-|-|-|-|-|-|-|
> ||PSNR↑|SSIM↑|LPIPS↓|PSNR↑|SSIM↑|LPIPS↓|PSNR↑|SSIM↑|LPIPS↓|PSNR↑|SSIM↑|LPIPS↓
> |FeMaSR|25.10|0.7858|0.3380|25.92|0.7991|0.3397|23.49|0.6921|0.4356|25.31|0.7450|0.4312
> |FeMaSR+SinCA|27.85|0.8178|0.3304|28.36|0.8295|0.3222|24.98|0.7094|0.4242|26.76|0.7635|0.4121
> |AdaCode|25.01|0.7733|0.3420|26.08|0.7898|0.3373|22.78|0.6762|0.4274|24.63|0.7333|0.4206
> |AdaCode+SinCA|27.13|0.8256|0.3133|27.91|0.8361|0.3098|24.31|0.6982|0.4164|26.08|0.7473|0.4121
>
> Once again, we sincerely thank you for affirming our work and providing constructive feedback and insightful suggestions, which have been invaluable in helping us improve the study.

---

### Official Review · Reviewer_JLcW · 2024-11-02

**Soundness:** 2
**Presentation:** 3
**Contribution:** 2
**Rating:** 3
**Confidence:** 4

**Summary:**

This paper proposes to span the discrete HQ codebook to continuous space and incorporate the LQ feature for VQ-based blind image restoration (BIR). Building upon the three side effects of VQ for code index prediction, the SinCA module is proposed. Experiments show the effectiveness of the proposed method.

**Strengths:**

1. This paper provides detailed analysis of the side effects of VQ-based code index prediction for BIR, including confined codebook representation capability, index prediction error, and LQ feature under exploitation.
2. Experiments show the effectiveness of the proposed SinCA module for blind image restoration.

**Weaknesses:**

1. This paper aims to point that the HQ codebook alone is insufficient for blind image restoration, however, many previous methods have proposed to utilize the LQ feature for quality and fidelity tradeoff, such as [CodeFormer; NeurIPS, 2022], it seems like we couldn't  accomplish the same balance in proposed SinCA framework currently, and the final result is totally deterministic and uncontrolled.
2. The paper argue that the representation capability of the discrete codebook is confined, is there any experiments to validate that when we span the codebook space to continuous space through the attention operation, could we obtain any benefit compared to the discrete code index? And I mean no LQ feature involved here.
3. The mutual attention adopted in SinCA is heavily utilized in literature, such as reference-based image/video manipulation, which may insufficient to support the contribution of the paper.
4. The illustration and notation of Fig. 3 is unclear, and should be rephrased.

**Questions:**

1. It is unclear what the different color mean in Fig. 3(a), and why multiple codebook marks interleaved with LQ feature.
2. In sec. 4.2, is that reasonable to formulate the GT indices for transformer prediction with nearest-neighbor feature matching, then why we need transformer, and the consequent conclusion are doubted.
3. It is suggested to provide a deeper analysis why Fig. 2(c) achieves extreme lower code prediction accuracy than Fig. 2(b), considering the codebook usage result in Fig. 2(a).

---

> ### Author Response · Authors · 2024-11-25
> **Response to The Weakness 1 (W1) And Weakness 2 (W2) of Reviewer JLcW**
>
> Thanks for the insightful feedback. We are grateful for your positive evaluation of the **detailed analysis of the side effects of discrete VQ** and the **effectiveness of the proposed SinCA module demonstrated with experiments**.
>
> **W1: Lacks controlling between quality and fidelity**.
>
> As illustrated in the Sec 5.3 of our paper, for different LQ images, SinCA adaptively balances the fidelity and quality by the self-part and cross-part in attention map. This adaptive mechanism ensures an implicit yet effective trade-off between using the HQ codebook and LQ features for BIR. Different CodeFormer, which employs an external parameter w for fidelity-quality balance, SinCA achieves this balance inherently.  The capability of SinCA on adaptive quality-fidelity trade-off has been validated in Sec. 6.2 (Tabs.1-2) in our main paper.
>
> We agree that the explicit quality-fidelity balance in CodeFormer is meaningful for BFR. However, our work aims for general replacement of VQ by SinCA for VQ-based BIR methods, not only for methods like CodeFormer. Many VQ-based BIR methods like FeMaSR, AdaCode, DAEFR, and RestorFormer do not have explicit quality-fidelity balance. To accomodate SinCA for general VQ-based BIR methods, we do not provide explicit balance between quality and fidelity in SinCA. In fact, it is not difficult to provide explicit quality-fidelity balance in SinCA. The key is how to strengthen the information from the self-part or the cross-part of the attention map in SinCA. This can be achieved by multiplying a weight hyper-parameter with the key matrix $\mathbf{K}_\mathbf{X}$ in Eqn. (4) of our main paper. We will consider it as a future work. Thanks  again for your constructive feedback.
>
> -----------------------
>
> **W2: Further exploration of benefits from continuous space**.
>
> To study the benefits from continuous space through attention, we conduct experiments where cross-attention (CA) replaces the discrete VQ process in the **Prior Learning** stage of FeMaSR and CodeFormer. Here, we freeze the encoder and decoder of the FeMaSR and CodeFormer in the **Prior Learning** Stage and only train the cross-attention module. In this setup, the representative space of codebook is expanded from **discrete** code items to **continuous** space through attention, which is an essential improvement. The results in **Tab. 1** validate that spanning codebook space into a continuous space indeed improve the representation capability of codebook with better reconstruction performance.
>
> Additionally, our ablation study has provided the image restoration results of four VQ-based BIR methods in Tab. 1 and their variants using CA in Tab. 4 of our main paper. For convenient readability, we summarize the above results in **Tab. 2**. Compared to the retrained baselines, all variants of four VQ-based BIR methods with a transformer using CA achieves clear improvements in most cases. This also validates the benefits of extending discrete codebook items to a continuous space for VQ-based BIR methods, further supporting our approach and motivation.
>
> **Tab.1 Reconstruction Results**
> | Method|Urban100|||DIV2K|||Method|CelebA|||
> |-|-|-|-|-|-|-|-|-|-|-|
> ||PSNR↑|SSIM↑|LPIPS↓|PSNR↑|SSIM↑|LPIPS↓||PSNR↑|SSIM↑|LPIPS↓
> |FeMaSR+VQ|20.52|0.6592|0.2345|23.98|0.6919|0.2496|CodeFormer+VQ|24.11|0.6784|0.1679
> |FeMaSR+CA|26.10|0.8667|0.1563|26.65|0.7853|0.2044|CodeFormer+CA|25.52|0.7137|0.1638
>
> **Tab.2 Restoration Results**
> | Method|Urban100|||DIV2K|||Method|CelebA|||
> |-|-|-|-|-|-|-|-|-|-|-|
> | |PSNR↑|SSIM↑|LPIPS↓|PSNR↑|SSIM↑|LPIPS↓||PSNR↑|SSIM↑|LPIPS↓
> |FeMaSR(VQ)|19.61|0.5607|0.4103|22.76 |0.6311|0.4129|CodeFormer(VQ)|22.66|0.6248|0.3100
> |FeMaSR+CA|20.08|0.5740|0.3957|23.22 |0.6424|0.3970|CodeFormer+CA|22.83|0.6310|0.3154
> |AdaCode(VQ)|19.47|0.5565|0.4124|22.55|0.6231|0.4102|DAEFR(VQ)|19.65|0.5456|0.3675
> |AdaCode+CA|19.53|0.5583|0.4247|22.67|0.6349|0.3654|DAEFR+CA|21.47|0.6001| 0.3770

---

> > ### Author Response · Authors · 2024-11-25
> > **Response to The Weakness 3 (W3) And Weakness 4 (W4) of Reviewer JLcW**
> >
> > **W3: SinCA lacks contribution**.
> >
> > 1. **Mutual attention is heavily utilized**.
> >
> > Here, we provide two key differences between SinCA and the mutual use of self-attention (SA) and cross-attention (CA) used in reference-based manipulation.
> >
> > (a) **Different motivation**. The mutual use of SA and CA is mainly from Stable Diffusion models for reference-based manipulation. The motivation of mutual SA and CA is to learn spatial information from noise map by SA and to edit the image from the guidance of reference by CA.
> >
> > Different from these motivations, our SinCA is the direct outcome from three observations. Through our first and second observations that VQ confines the representation space of HQ codebook and VQ is error-prone, we propose the necessity to replace discrete VQ by continuous feature transformation.
> >
> > The insights from our third observation motivates us to integrate LQ feature and HQ codebook for adaptive balance between restoration fidelity and quality. Thus, we propose SinCA to simultaneously perform SA and CA in a unified module, which is implemented by performing cross-attention between LQ feature and input-augmented codebook.
> >
> >
> > (b) **Different structure**. In Stable Diffusion models for image editing, SA and CA are operated sequentially as independent modules. In contrast, our SinCA performs cross-attention between the LQ feature and input-augmented codebook. This can be viewed as simultaneous performing SA and CA in a unified module, ensuring adaptive trade-off between the restoration fidelity in SA and quality in CA of our SinCA in VQ-based BIR methods. This is very different from the sequential operation of SA and CA in previous reference-based image manipulation methods.
> >
> >
> > One principal contribution of our work lies in our thorough analysis and validation of an improvement direction for general VQ-based BIR methods: replacing discrete VQ with continuous feature transformation. To this end, we proposed a ''simple but effective'' SinCA module to enhance the restoration fidelity while preserving the quality of VQ-based BIR methods. We are very appreciated that our work has been corroborated by the positive assessments of **Reviewer WGGn** and **Reviewer r6SX**. They endorse that the ''detailed'' observations, coupled with the implementation of our SinCA, offer ''**a compelling solution to overcome VQ's limitation**'', while providing ''**a different perspective**'' for future research. Their feedback substantiates the unique aspects and innovative potential of our approach.
> >
> > -----------------------
> > **W4: Clarify illustration of Fig. 3**.
> >
> > We've rephrased the illustration of Fig. 3 as follows.
> >
> > **T-SNE visualization of VQ process** in FeMaSR (a) and CodeFormer (b). Different code items in HQ codebook are marked by "☆" in different colors. The color of  LQ feature vector marked by "◯'' or  the HQ feature vector marked by "△'' is the same with the codebook item they select in VQ process. Gray dashed lines "- -'' connects the LQ feature vector "◯'' and its selected codebook item "☆"  (“VQ Process”). Red dash lines "- -'' connects the LQ feature vector  "◯''  and the codebook item "☆" selected by the corresponding HQ feature vector using nearest-neighbor (NN) feature matching in the **Prior Learning** Stage (“GT Selection”).
> > (a) NN feature matching on LQ feature vectors are inconsistent with “GT Selection”. For a given LQ feature vector, the codebook item selected by NN feature matching (gray dash line, "- -'') is quite different from the corresponding "GT Selection'' (red dashed line, "- -'').
> >  (b) Transformer for code index prediction is not robust to image degradation. In a degraded LQ image, an "LQ Eye'' patch looks like skin area and selects the code item represented by many "HQ Skin'' patches. The regions marked by purple dashed box and orange dashed box are enlarged for better view ("Enlarge'').
> >  (c) Prediction accuracy of code indices by transformer and SSIM results achieved by CodeFormer using "LQ Indices'' predicted on LQ images, "HQ Indices'' predicted on HQ images, or "GT Indices'' defined in Sec.4.2.

---

> > > ### Author Response · Authors · 2024-11-25
> > > **Response to Questions 1-3 (Q1-3) of Reviewer JLcW**
> > >
> > > **Q1: Clarify marks in Fig. 3(a)**.
> > >
> > > We'd like to clarify this aspect. Fig. 3(a) shows the t-SNE visualization of the VQ process in FeMaSR. We project both the LQ feature vectors and the codebook items into a 2-D space using t-SNE. In this figure, each codebook item is represented by a five-pointed star "☆" with a distinct color, and the LQ feature vectors "◯'' are filled with the same color as the codebook item they select. The interleaving of multiple codebook items with LQ feature vectors suggests that these LQ feature vectors are close to some codebook items in the projected 2D-space, which means that the compressed content of the LQ feature vectors and HQ codebook items are similar. This does not indicate that LQ feature vectors and HQ codebook items are interleaved in high-dimensional vector space.
> > >
> > > -------------
> > >
> > > **Q2: The rationale of choosing GT indices**.
> > >
> > > The choice of GT indices follows the definition established in prior VQ-based BIR methods such as CodeFormer and DAEFR. As mentioned in the Sec.4.2 (Page 5, line 247) of our main paper, these works define GT indices as the indices selected by the HQ feature extracted from input HQ image by the encoder in the **Prior Learning** stage, and the selection is performed by nearest neighborhood feature matching. These GT indices are used as ground-truth to supervise the learning of transformer for VQ using the cross-entropy loss.
> > >
> > > Since HQ images are in similar quality to high-quality codebook items, using nearest-neighbor feature matching provides a reliable way to select GT indices. Moreover, as shown in Fig. 3(c), using GT indices brings better restoration performance than using predicted indices, which further validates the reasonableness of this choice.
> > >
> > > ---------------
> > >
> > > **Q3: Deeper analysis of low accuracy in BSR**.
> > >
> > > As shown in Fig. 2(a) and Fig. 2(b), the codebook usage ratios in BSR methods of FeMaSR and AdaCode are significantly lower than those of BFR methods of CodeFormer and DAEFR. For instance, the codebook usage ratio of FeMaSR is only 3.32% (34/1024) while DAEFR uses 100% (1024/1024) codebook for image restoration. The lower codebook usage ratio in BSR reduces the difficulty of code prediction task due to fewer choices of available code items. For FeMaSR, 48 code items are active in the **Prior Learning** stage while the **Image Restoration** stage further narrows this scope to a subset of 34 code items. This reduced diversity in codebook usage contributes to the relatively low accuracy observed in Fig. 2(c). We will add the discussion on this issue into our revised submission.
> > >
> > > --------
> > > We appreciate your constructive comments, and hope that our responses could address your concerns on quality-fidelity trade-off, contributions, and presentation, etc. Considering the contributions of our work on general improvements of VQ-based BIR methods and more detailed clarification in our response, we kindly request you to consider revising your score to recommend acceptance. Your thoughtful review and constructive suggestions have been invaluable.

---

> > > > ### Comment · Reviewer_JLcW · 2024-11-27
> > > > **Official Comment by Reviewer JLcW**
> > > >
> > > > Thanks for the authors' detailed response. However, the main concerns are remain unsolved and basically for the core contribution of this work.
> > > >
> > > > As presented in W1, "This paper aims to point that the HQ codebook alone is insufficient for blind image restoration, however, many previous methods have proposed to utilize the LQ feature for quality and fidelity tradeoff", which is basically the same idea as this work but different implementation methods.
> > > >
> > > > The main method for incorporating the Lq features is to utilize the mutual attention in SinCA, which is also heavily utilized in literatures for concurrent self-attention and cross-attention, such as reference-based image/video manipulation, like [Animate Anyone; CVPR2024, MagicAnimate; CVPR2024], and may insufficient to support the contribution of this paper.
> > > >
> > > > The response to Q3 is somewhat inconsistency with the observation in Fig. 2. Actually, what I observed is that the reduced diversity in codebook usage contributes to the relatively higher logits prediction accuracy.
> > > >
> > > > Therefore, I will maintain my score based on the above concerns.

---

> ### Author Response · Authors · 2024-11-29
> **Response to concern that HQ codebook alone is insufficient for BIR and previous work also using LQ feature**
>
> Thank you for taking the time to consider our response. We would like to address the remaining concerns.
>
> 1. **HQ codebook alone is insufficient for BIR and previous work also using LQ feature**
>
> Our work does not aim to point out that HQ codebook alone is insufficient for BIR. Instead, our work focuses on tackling the limitations of VQ for general BIR methods using VQ: for W1, our work claims that **the representation capability of HQ codebook is confined by VQ** (our **Observation 1**). This point has be validated by replacing VQ by cross-attention (CA) between LQ feature and HQ codebook: the results in Tabs. 1 and 2 of the response to **W2** show that continuous CA outperforms discrete VQ on both image reconstruction and restoration.
>
> As presented in Sec. 4.2 of our paper (Page 5, line 267), we agree that **previous work used LQ feature for quality-fidelity trade-off**, but the feature fusion is implemented **after VQ**. However, these methods still suffer from **the limited representative capability and under-utilization of LQ feature in the VQ process itself, and they do not directly address the root cause within the VQ process**. On the contrary, our work reveals that LQ feature is under-utilized in VQ (our **Observation 3**). Thus, we propose to augment HQ codebook with LQ feature and replace discrete VQ process by continuous feature attention **in SinCA (as a better alternative of VQ)** for better quality-fidelity balance. **This not only directly addresses the under-utilization issue of LQ feature in the VQ process, but also expands the codebook’s representational capacity to effectively tackling the key limitations of VQ we illustrated in our three observations**.
>
> On **core contribution**, we clarify that our work is motivated by three key observations on the limitations of VQ process in general VQ-based BIR methods. To address these limitations for better BIR performance, our **core contribution** is to replace discrete VQ by continuous feature learning, which avoids these limitations and is more reasonable for BIR task. Our SinCA is a "simple but effective" exploration in this direction. SinCA is able to perform self-attention on LQ feature itself for restoration fidelity and cross-attention between LQ feature and HQ codebook for restoration quality, and meanwhile achieving adaptive balance between restoration quality and fidelity. To achieve these goals, we propose SinCA to perform cross-attention between input-augmented codebook and LQ feature of input LQ image.
>
> While CodeFormer explicitly attempts to achieve trade-off between fidelity and quality, other VQ-based BIR methods like FeMaSR and AdaCode do not achieve this trade-off but employ feature fusion for performance enhancement. The results of FeMaSR and AdaCode are deterministic, whereas SinCA achieves input-adaptive enhancement on both quality and fidelity by integrating LQ information into SinCA for feature transformation. Moreover, DAEFR does not have a fusion module to connect encoder and decoder. **Our SinCA replaces VQ and thus can be applied to general VQ-based BIR methods, whether or not a feature fusion module is used after VQ. **This makes our SinCA highly versatile and compatible with existing methods that already incorporate fusion modules, providing meaningful improvements on both quality and fidelity.
>
> Furthermore, as referred by the reviewer in W1, CodeFormer explicitly achieves the fidelity-quality trade-off through an LQ feature fusion module. When LQ images are severely degraded, removing the fusion module can lead to higher-quality restored images but at the cost of fidelity. To demonstrate the impact of our approach, we compare "CodeFormer+SinCA" with CodeFormer both with and without the fusion module in **Figs. 3-5** of the **[anonymous link](https://anonymous.4open.science/r/ICLR2025-FC45/)**  (File name: 'Visual\_Results\_1.pdf'). Clicking "view raw"  can provide a better view. Notably, **the visual results of "CodeFormer+SinCA w/o fusion" are competitive with, or even surpass, those of "CodeFormer w/o fusion" and "CodeFormer w fusion" (vanilla CodeFormer)**. It is important to emphasize that our goal is not to replace or remove the fusion modules in previous works, but rather to demonstrate that by replacing the discrete VQ process with our proposed SinCA, we can address the performance bottleneck of the VQ process and unlock new potential for improving blind image restoration.

---

> ### Author Response · Authors · 2024-11-29
> **Response to Concern about Concurrent Use of SA And CA in Previous Work and Q3**
>
> 2. **Concurrent use of self-attention and cross-attention**.
>
> Thank you for bringing the referred papers to our attention. Our work **is not motivated by these references, but by our observations regarding the VQ process in VQ-based BIR methods**. After carefully reviewing the literature, we believe that the novelty of these works does not diminish the contributions of our paper. **Our work aims to study the limitations of VQ for BIR and give a novel solution to replace the VQ process, extending the codebook representative space and adaptively preseving the fidelity while enhancing the quality**. SinCA is one solution we propose to achieve our goal.
>
> The referred papers, while useful in their respective contexts, do not address the limitations of VQ process, which instead is the goal of our work. The use of self-attention and cross-attention in those works **is not relevant to the limitations of the discrete VQ process in VQ-based BIR methods**.
>
> For example, Animate Anyone [1] uses **spatial attention** to extract spatial information, **cross-attention** for useful guidance from reference images, and **temporal attention** to capture dependencies during denoising steps for character animation. MagicAnimate [2] employs mutual **self-attention** and **cross-attention** to ensure temporal consistency while incorporating guidance from the reference image during denoising steps for character animation task. **These two approaches focus on different objectives and do not tackle the limitations of the VQ process in BIR methods**.
>
> Different from these two work, our work introduces a general replacement of the VQ process in VQ-based BIR methods, achieving promising improvements of these methods on BIR tasks. Since the contributions of our work are very different from those of the above-referred works, we believe that our work is novel and meaningful in advancing the field of BIR, especially the popular branch of BIR methods developed under the VQ framework (e.g., VQVAE, VQGAN, or others).
>
> ---
>
> **Error in the response to Q3**.
>
> Thanks for pointing out this. It's a slip of a pen. We want to say that the reduced diversity in codebook usage contributes to the relatively higher accuracy.  We apologize for the oversight and appreciate your careful attention to detail. We'd like to correct the response to Q3 as follows:
>
> As shown in Fig. 2(a) and Fig. 2(b), the codebook usage ratios in BSR methods of FeMaSR and AdaCode are significantly lower than those of BFR methods of CodeFormer and DAEFR. For instance, the codebook usage ratio of FeMaSR is only $3.32\%$ (34/1024) while DAEFR uses $100\%$ (1024/1024) codebook for image restoration. The lower codebook usage ratio in BSR reduces the difficulty of code prediction task due to fewer choices of available code items. For FeMaSR, 48 code items are active in the **Prior Learning** stage while the **Image Restoration** stage further narrows this scope to a subset of 34 code items. This reduced diversity in codebook usage contributes to the relatively higher accuracy observed in Fig. 2(c).
>
> ---
> We hope our responses have effectively addressed your concerns regarding the contributions of SinCA. Considering the strengths of our work, we kindly request you to reconsider your rating and recommend acceptance. Your insightful review and constructive feedback have been very helpful in improving our paper.
>
> ---
>
> >  [1] Xu, Z., “MagicAnimate: Temporally Consistent Human Image Animation using Diffusion Model”, CVPR, 2024.
>
> >  [2]Hu, L., Gao, X., Zhang, P., Sun, K., Zhang, B., and Bo, L., “Animate Anyone: Consistent and Controllable Image-to-Video Synthesis for Character Animation”, arXiv e-prints, Art. no. arXiv:2311.17117, 2023.

---

### Official Review · Reviewer_tazg · 2024-11-02

**Soundness:** 2
**Presentation:** 3
**Contribution:** 2
**Rating:** 3
**Confidence:** 5

**Summary:**

This paper theoretically analyzes the shortcomings of recent VQ algorithm framework in BIR tasks, and aims to illustrate that the one-hot form of code matching will cause limitations in the expressive ability of the codebook, a high matching error rate, and underestimation of the role of low-quality features, resulting in a decrease in recovery performance. Therefore, the paper proposes the Self-in-Cross-Attention (SinCA) module, which replaces discrete feature matching with continuous feature transformation, and introduces low-quality feature into cross attention by the input-enhanced codebook, achieveing better quantitative and qualitative performance on both BSR and BFR tasks.

**Strengths:**

1. The paper conducts a large number of experiments to demonstrate the validity of the three observations of VQ on BIR task. These three observations significantly exist in the existing VQ-based methods, resulting in poor performance.

2. A self-in-cross-attention module is proposed to tackle the issue of the discrete VQ process in code index prediction by the continuous feature transformation.  The idea about using LQ image feature and HQ codebook in cross attetion is simple but effective.

**Weaknesses:**

Although the author's demonstration of the shortcomings of VQ discrete matching is relatively sufficient, it is not novel, and there are also some problems in some places.

1. Originality is somewhat lacking: in Section 4.1,  the codeword utilization rates of the FeMaSR and AdaCode algorithms in the BSR task are low, and  Section 4.2 indicates that the encoding prediction accuracy of the CodeFormer and DAEFR algorithms in the BFR task is low. These two key issues have actually been described in previous works and is not the first time they have been brought up by the author. Many works have proposed solutions to the codebook collapse, and the problem of low prediction accuracy has actually been depicted in related charts in CodeFormer (Fig 6 in ref[1], the accuracy of Transformer encoding prediction is greater than that of the nearest neighbor, but still relatively low).

       [1] Shangchen Zhou, Kelvin Chan, Chongyi Li, and Chen Change Loy. Towards robust blind face restoration with codebook lookup transformer. Adv. Neural Inform. Process. Syst., 35:30599– 30611, 2022.

2. experiments are insufficient： Could you demonstrate that  the proposed SinCA module can ensure that the image quality does not degrade while enhancing fidelity (visualization + more comprehensive subjective + objective indicator data proof), which has no connection with the feature fusion module. In Section 4.3, by removing the feature fusion modules of FeMaSR, AdaCode, and CodeFormer, it is discovered that the PSNR, SSIM, and LPIPS indicators decline. This is used to prove the significance of the information in the input low-quality images (whether it is the multi-scale features in the Encoder or the low-quality compressed features before feature matching). A crucial point in the BFR task is actually overlooked, namely, maintaining the balance between fidelity and quality. For instance, CodeFormer achieves this through the hyperparameter w. If w is 0, then the CFT feature fusion module does not function, ensuring that no low-quality information enters, thereby enhancing the image quality compared to the situation where CFT is effective. The decrease in subjective indicators in this case is evident, but the objective indicators are likely to improve, especially in the case of severely degraded input.

3. The contibution is not enough, because the paper mainly includes several obsevations with slightly lese novelty and propose a simple attention module SinCA.

**Questions:**

Could you provide a more comprehensive experiment to show that introducing low-quality features can achieve high-quality restoration results on heavy degradation restoration tasks (such as the WebPhoto/Wider datasets of human faces) while improving fidelity on more real-world datasets for BFR tasks other than CelebAChild.

---

> ### Author Response · Authors · 2024-11-24
> **Response to the W1 of Reviewer tazg**
>
> Thank you for your detailed review and constructive feedback on our work. We are grateful for your acknowledgment of the **sufficient demonstration on the shortcomings of discrete VQ**, proposed **SinCA achieving better quantitative and qualitative performance** on both BSR and BFR tasks, as well as your positive remarks on our **simple but effective idea**.
>
> **W1: Originality is somewhat lacking**.
>
> **Low codebook usage and encoding prediction accuracy are already described in previous work**. Briefly, these are not key observations of our work. **The low codebook usage rate is not what we observe in our work**. Though mentioned in previous works, the issue of low codebook usage rate is not always right. In fact, the issue of low codebook usage rate does not exist in CodeFormer and DAEFR for BFR task. This contradiction indicates that usage rate of codebook in VQ-based BIR methods is still an open problem that need experimental supports, which makes it meaningful to performexperiments to illustrate  principles behind our observations. We will explain each point in details as follows.
>
> (a) **On codebook usage**.  Our first observation is that discrete VQ confines representation capability of HQ codebook to a finite set of code items, which is seldom mentioned in previous works. To study benefits of extending the discrete codebook to continuous space, inspired by **Reviewer JLcW**, we perform experiments to compare the performance of FeMaSR/CodeFormer using VQ or cross-attention (CA) on BIR. We summarize the results of FeMaSR and CodeFormer by replacing the discrete VQ with the continuous CA on both the image reconstruction task in the **Prior Learning** stage in **Tab. 1** (with fixed encoder and decoder) and the image restoration task in the **Image Restoration** stage in **Tab. 2** (with fixed decoder). It can be seen that CA improves both PSNR and SSIM results with 0.17$\sim$5.58dB and 0.0062$\sim$0.2075, respectively. This validates that replacing the discrete VQ with the continuous CA for feature learning significantly enhances the reconstruction and restoration performance of VQ-based BIR methods. We mentioned in our paper that this issue ''will be amplified by low codebook usage rates mentioned in previous works''. But this issue does not occur in CodeFormer and DAEFR developed for BFR. As illustrated in Fig. 2 (a) in our **Observation 1** (Section 4.1 of our main paper), the rates of codebook usage in CodeFormer and DAEFR are 98.73\% and 100\%, respectively.
>
> (b) Our second observation is that VQ is error-prone on LQ features. Even though mentioned in CodeFormer, it is still not an obvious conclusion on other VQ-based BIR methods. We thoroughly analyze the key reasons for this issue and find that the low accuracy of index prediction degrades the performance of VQ-based BIR methods. Besides, we also point out that higher prediction accuracy brings better image restoration performance (Fig. 3 (c) in our main paper). From these insights, we understand that it is challenging to enhance prediction accuracy given the inherent degradations. Consequently, we have opted to replace the error-prone index prediction with our SinCA, which significantly improves restoration performance.
>
> These two observations demonstrate that VQ is a two-sided coin with clear rewards and punishments for VQ-based BIR methods. This motivate us to replace the discrete VQ operation by a continuous feature transformation module for BIR. Besides, the third observation showing the importance of LQ feature for BIR motivates us to integrate the self-attention of LQ feature in the cross-attention between the LQ feature and the HQ codebook, and develop our Self-in-Cross attention (SinCA) module to fully exploit the useful information in LQ feature and HQ codebook. The objective of our work is not to address the issue of ''codebook collapse'', where only a small part of HQ codebook is used during the VQ operation.
>
> Our work provides the necessity to replace VQ by continuous feature learning in VQ-based BIR methods. Our SinCA is ''simple but effective'' for VQ-based BIR and a meaningful originality for low-level vision.
>
> **Tab.1 Reconstruction Results**
> | Method|Urban100|||DIV2K|||Method|CelebA|||
> |-|-|-|-|-|-|-|-|-|-|-|
> ||PSNR↑|SSIM↑|LPIPS↓|PSNR↑|SSIM↑|LPIPS↓||PSNR↑|SSIM↑|LPIPS↓
> |FeMaSR(VQ)|20.52|0.6592|0.2345|23.98|0.6919|0.2496|CodeFormer(VQ)|24.11|0.6784|0.1679
> |FeMaSR+CA|26.10|0.8667|0.1563|26.65|0.7853|0.2044|CodeFormer+CA|25.52|0.7137|0.1638
>
> **Tab.2 Restoration Results**
> | Method|Urban100|||DIV2K|||Method|CelebA|||
> |-|-|-|-|-|-|-|-|-|-|-|
> | |PSNR↑|SSIM↑|LPIPS↓|PSNR↑|SSIM↑|LPIPS↓||PSNR↑|SSIM↑|LPIPS↓
> |FeMaSR(VQ)|19.61|0.5607|0.4103|22.76 |0.6311|0.4129|CodeFormer(VQ)|22.66|0.6248|0.3100
> |FeMaSR+CA|20.08|0.5740|0.3957|23.22 |0.6424|0.3970|CodeFormer+CA|22.83|0.6310|0.3154
> |AdaCode(VQ)|19.47|0.5565|0.4124|22.55|0.6231|0.4102|DAEFR(VQ)|19.65|0.5456|0.3675
> |AdaCode+CA|19.53|0.5583|0.4247|22.67|0.6349|0.3654|DAEFR+CA|21.47|0.6001| 0.3770

---

> ### Author Response · Authors · 2024-11-25
> **Response to the W2 of Reviewer tazg**
>
> **W2: insufficient experiments.**
>
> 1.**Whether SinCA preserve quality while enhancing fidelity without feature fusion** .
> To study this, we train FeMaSR and CodeFormer using SinCA by removing the corresponding feature fusion module. The quantitative results are summarized in **Tabs. 3-4**. Qualitative comparisons are provided in **Figs. 1-5** of our provided **[anonymous link](https://anonymous.4open.science/r/ICLR2025-FC45/) (File name: 'Visual_Results_1.pdf')**. One can see that without feature fusion, FeMaSR and CodeFormer using SinCA preserve image quality while enhancing fidelity on BIR tasks.
>
> We agree that human subjects are the ultimate judge of image quality, and did subjective user study to evaluate the restoration quality and fidelity of "CodeFormer w/o fusion'' and ''CodeFormer+SinCA w/o fusion''. This study involves 32 participants, and each participant is asked to do preference evaluation on restored images of 25 LQ images randomly selected from Wider, WebPhoto, and LFW. In each trial, each participant was presented with a set of three images: one LQ input for the reference of restoration fidelity, the image restored by ''CodeFormer w/o fusion'', and the image restored by ''CodeFormer+SinCA w/o fusion''. In each set of presentation, the first image is LQ input while the order of two restored images is randomly shuffled. Results of user preference voting are listed in **Tab. 5**, which show that''CodeFormer+SinCA w/o fusion'' got 73.75% of total 800 votes.
>
> The results validate that SinCA with adaptive quality-fidelity balance indeed preserves image quality while enhancing the fidelity of VQ-based BIR methods like FeMaSR and CodeFormer.
>
> 2.**Overlooking the fidelity-quality balance in BFR**.
> As illustrated in the Sec 5.3 of our paper, for different LQ images, SinCA adaptively balances the fidelity and quality by the self-part and cross-part in attention map. The adaptive balance in SinCA ensures implicit yet effective trade-off between using HQ codebook and LQ features for BIR. This is different from CodeFormer that uses an external parameter w for fidelity-quality balance. The capability of SinCA on adaptive quality-fidelity trade-off has been validated in Sec. 6.2 (Tabs.1-2) in our main paper.
>
> To further support SinCA on BFR, we do more experiments of CodeFormer on real-world face datasets LFW, Wider, and WebPhoto. As noted by the reviewer, CodeFormer achieves enhanced quality when w=0. Thus, we compare "CodeFormer+SinCA'' with CodeFormer when w=0 (w/o fusion). In **Tab.4**, the results on the fidelity metrics of PSNR and SSIM computed on input LQ image and the quality metrics of NIQE and Perceptual Index (PI) show that ``CodeFormer+SinCA w/o fusion'' achieves adaptive balance on quality and fidelity. The visual comparisons are provided in **Figs. 3-4** of the **[anonymous link](https://anonymous.4open.science/r/ICLR2025-FC45/)  (File name: 'Visual_Results_1.pdf')**. We recommend the reviewer to click "view raw'' for a better view. Notably, the visual results of "CodeFormer+SinCA w/o fusion'' are competitive with or even surpass those of "CodeFormer w/o fusion'' (w=0) and "CodeFormer w fusion'' (vanilla CodeFormer).
>
> We agree that the explicit quality-fidelity balance in CodeFormer is meaningful for BFR. However, our work aims for general replacement of VQ by SinCA for VQ-based BIR methods, not only for methods like CodeFormer. Many VQ-based BIR methods like FeMaSR, AdaCode, DAEFR, and RestorFormer do not have explicit quality-fidelity balance. To accomodate SinCA for general VQ-based BIR methods, we do not provide explicit balance between quality and fidelity in SinCA. In fact, it is not difficult to provide explicit quality-fidelity balance in SinCA. The key is how to strengthen the information from the self-part or the cross-part of the attention map in SinCA. This can be achieved by multiplying a weight hyper-parameter with the key matrix $\mathbf{K}_\mathbf{X}$ in Eqn. (4) of our main paper. We will consider it as a future work. Thank you very much for your constructive suggestion.
>
> **Tab 3 Results of FeMaSR w/o fusion**
> |Method|Urban100|||DIV2K|||B100|||Manga109|||
> |-|-|-|-|-|-|-|-|-|-|-|-|-|
> ||PSNR|SSIM|LPIPS|PSNR|SSIM|LPIPS|PSNR|SSIM|LPIPS|PSNR|SSIM|LPIPS
> |using VQ|19.08|0.5287|0.4418|22.17|0.6122|0.4360|21.31|0.4965|0.4631|20.40|0.6764|0.3825
> |using SinCA|20.22|0.5555|0.4210|24.09|0.6564|0.4119|22.37|0.5322|0.4479|20.82|0.6983|0.3645
>
> **Tab 4 Results of CodeFormer w/o fusion**
> |Method|CelebA|||Webphoto||||Wider||||LFW||||
> |-|-|-|-|-|-|-|-|-|-|-|-|-|-|-|-|
> ||PSNR|SSIM|LPIPS|LR PSNR|LR SSIM|NIQE↓|PI↓|LR PSNR|LR SSIM|NIQE↓|PI↓|LR PSNR|LR SSIM|NIQE↓|PI↓|LR PSNR|LR SSIM|
> |using VQ|21.77|0.5963|0.3250|26.01|0.7563|4.6258|3.93|24.12|0.6912|4.0871|3.12|25.20|0.7334|4.2845|3.25
> |using SinCA|22.31|0.6051|0.3367|26.34|0.7555|4.5938|3.85|24.36|0.6945|4.1245|3.12|25.71|0.7369|4.2782|3.20
>
>
> **Tab 5 User study results**
> |w/o fusion|CodeFormer|CodeFormer +SinCA
> |-|-|-|
> |Preferrence|26.25%|73.75%

---

> > ### Author Response · Authors · 2024-11-25
> > **Response to the W3 and Q1 of Reviewer tazg**
> >
> > **W3: Not enough contributions.**
> >
> > One principal contribution of our work lies in our thorough analysis and validation of an improvement direction for general VQ-based BIR methods: replacing discrete VQ with continuous feature transformation. To this end, we proposed a ''simple but effective'' SinCA module to enhance the restoration fidelity while preserving the quality of VQ-based BIR methods. We are very appreciated that our work has been corroborated by the positive assessments of **Reviewer WGGn** and **Reviewer r6SX**. They endorse that the ''detailed'' observations, coupled with the implementation of our SinCA, offer ''**a compelling solution to overcome VQ's limitation**'', while providing ''**a different perspective**'' for future research. Their feedback substantiates the unique aspects and innovative potential of our approach.
> >
> > Specifically, the **observation 1** indicates that discrete VQ confines the representation of HQ codebook to a finite set of code items. This non-trivial conclusion motivates us to extend the representation into a continuous space of HQ codebook and achieve the performance improvements when replacing VQ with continuous CA (as shown in Tab. 1 and Tab. 2). The **observation 2** identifies that the VQ process is error-prone on LQ features. Although CodeFormer acknowledges the issue of low prediction accuracy that degrade the restoration performance, we investigate this aspect in other VQ-based BIR methods and find that higher prediction accuracy can improve the restoration performance. The **observation 3** points out that the importance of LQ features is undervalued in VQ process. This aspect is inherently overlooked by the VQ process in previous VQ-based BIR methods, though somewhat alleviated by further fusion of LQ and HQ features in decoder. To well-utilize the LQ information, it is necessary to replace VQ by continuous feature learning for the enhancement of final restoration performance.
> >
> > Based on these analysis, it is naturally formulate our SinCA module as an effective replacement of discrete VQ to perform cross-attention between input LQ feature and input-augmented codebook. By exploiting the self-expressiveness of LQ feature and the correlation between LQ feature and HQ codebook, it is able to achieve adaptive quality-fidelity balance on restoration results. We are very appreciated that our SinCA is ''a simple but effective'' module to yield clear improvements on VQ based BIR methods.
> >
> > **Q1: More results on real-world datasets for BFR**.
> >
> > In **Figs. 6-8** of the attached PDF file (File name:'Visual\_Resulst\_2.pdf') in our **[anonymous link](https://anonymous.4open.science/r/ICLR2025-FC45/)**, we provide visual comparisons on the real-world datasets such as WebPhoto-Test, WIDER-Test and LFW-Test, respectively. It can be found that the proposed SinCA successfully preserves the visual details of human face such as the hair, eyes, and lips. We also provide the numerical results among the three real-world datasets in Tab. 6, together with the result of subjective user study provided in **Tab. 6**. These results validate that introducing LQ feature in our SinCA achieves high-quality restoration while improving fidelity on heavy degradations in real-world image restoration tasks.
> >
> > **Tab. 6 Results of BFR on real-world datasets**
> > | Method|Webphoto||||Wider||||LFW||||
> > |-|-|-|-|-|-|-|-|-|-|-|-|-|
> > || LR PSNR↑| LR SSIM↑| NIQE↓| PI↓ | LR PSNR↑| LR SSIM↑| NIQE↓ | PI↓ |LR PSNR↑| LR SSIM↑| NIQE↓ | PI↓ | LR PSNR↑| LR SSIM↑| NIQE↓ | PI↓
> > |CodeFormer| 26.01|0.7563  |4.6258 | 3.93| 24.12	| 0.6912| 4.0871 | 3.12| 25.20| 0.7334 | 4.2845 | 3.25
> > |CodeFormer + SinCA | 26.34| 0.7555 | 4.5938 | 3.86|  24.36| 0.6945 | 4.1245 | 3.12| 25.71| 0.7369| 4.2782 | 3.20
> > |DAEFR | 24.31 | 0.7469 | 4.1751 | 3.78 | 22.30 | 0.6846 | 3.8955 |	3.29| 24.46 |0.7199 | 3.7471 | 3.09
> > |DAEFR + SinCA | 25.69 | 0.7555 | 4.2398 | 3.92| 23.76 | 0.7091 |3.93 | 3.4238| 25.08 | 0.7210 |3.6521 |2.98
> >
> > We hope our responses address your concerns and clarify the originality, effectiveness, and robustness of SinCA. Given the strengths of our work, we kindly request you to consider raising your score to recommend acceptance. Your thoughtful review and constructive suggestions have been invaluable, and we are committed to further refining the paper based on this feedback.

---

> > > ### Comment · Reviewer_tazg · 2024-11-26
> > >
> > > I would like to express my gratitude to the authors for their comprehensive and meticulous response as well as for undertaking extensive revisions to the manuscript. The incorporation of an additional case study based on a greater variety of datasets, namely WebPhoto-Test, WIDER-Test, and LFW-Test, is a highly valuable and welcome enhancement. I have meticulously examined all the comments provided by the other reviewers and the corresponding responses from the authors. Overall, my initial evaluation and rating of the paper remain unchanged.

---

> > > > ### Author Response · Authors · 2024-11-29
> > > > **Response to reviewer's gratitude**
> > > >
> > > > Thanks for your gratitude to our response. We have updated our reply to other reviewers, providing further clarifications. We believe that our work provides meaningful insights for VQ-based BIR methods. We sincerely hope to gain your support to our work.

---

> > > ### Public Comment · ~Jingkai_Wang2 · 2025-02-24
> > > **Question Regarding the Use of LR PSNR and LR SSIM**
> > >
> > > Dear Authors,
> > >
> > > I hope this message finds you well. I noticed in your rebuttal that you used LR calculation for PSNR and SSIM in the context of real-world images. I was wondering if this approach is commonly accepted or if there might be any specific reasoning behind it. Would you be able to share any additional references that utilize LR images and predicted images for full-reference IQA?
> > >
> > > Thank you for your time and consideration.

---

> > > > ### Public Comment · ~Tianyi_Xu4 · 2025-02-24
> > > > **Response to The Question  Regarding the Use of LR PSNR and LR SSIM**
> > > >
> > > > Thanks  for your interests and attention.
> > > >
> > > > Calculating the PSNR and SSIM between the LR input and the restored image can assess how closely the output matches the original LR input, providing insights into the fidelity of the restoration process, especially in situations where a generative prior is used for image restoration. Here we provide some referece papers.
> > > > 1. Park S H, Moon Y S, Cho N I. Perception-oriented single image super-resolution using optimal objective estimation[C]//Proceedings of the IEEE/CVF conference on computer vision and pattern recognition. 2023: 1725-1735.
> > > > 2. Lugmayr A, Danelljan M, Van Gool L, et al. Srflow: Learning the super-resolution space with normalizing flow[C]//Computer vision–ECCV 2020: 16th European conference, glasgow, UK, August 23–28, 2020, proceedings, part v 16. Springer International Publishing, 2020: 715-732.
> > > > 3. A. Lugmayr et al., "NTIRE 2022 Challenge on Learning the Super-Resolution Space," 2022 IEEE/CVF Conference on Computer Vision and Pattern Recognition Workshops (CVPRW), New Orleans, LA, USA, 2022, pp. 785-796, doi: 10.1109/CVPRW56347.2022.00094.
> > > > 4.Li-Yuan Tsao, Yi-Chen Lo, Chia-Che Chang, Hao-Wei Chen, Roy Tseng, Chien Feng, and Chun-Yi Lee. Boosting Flow-based Generative Super-Resolution Models via Learned Prior. Proceedings of the IEEE/CVF Conference on Computer Vision and Pattern Recognition (CVPR), June 2024, pp. 26005–26015.

---

### Meta-Review · Area_Chair_i4xR · 2024-12-18

**Metareview:**

This paper proposes an effective SinCA to improve the representational capability of the HQ codebook by leveraging the self-expressiveness property of input LQ image. The provided results show the effectiveness of the proposed method.

The paper receives reviews with significant divergent ratings.  Reviewer tazg and Reviewer JLcW mainly pointed out that the paper is not novel. In addition, the experimental evaluations are not sufficient as pointed out by Reviewer tazg.  Moreover, Reviewer r6SX mainly criticizes the complexity and generalization ability of the proposed method.

After rebuttal, the authors partly solve the concerns of Reviewer tazg and Reviewer r6SX. However, the concerns about the limited novelty still remains.

Based on the recommendations of reviewers, the paper is not accepted.

**Additional Comments On Reviewer Discussion:**

In the discussion stage,  Reviewer tazg and Reviewer JLcW still remain their ratings due to the limited novelty.

---

### Decision · Program_Chairs · 2025-01-22

Reject